DOI: 10.1038/s41467-018-06697-x · **OPEN**

# The H3K9 methyltransferase SETDB1 maintains female identity in *Drosophila* germ cells

Anne E. Smolko [1], Laura Shapiro-Kulnane[1] & Helen K. Salz [1]

The preservation of germ cell sexual identity is essential for gametogenesis. Here we show that H3K9me3-mediated gene silencing is integral to female fate maintenance in *Drosophila* germ cells. Germ cell specific loss of the H3K9me3 pathway members, the H3K9 methyltransferase SETDB1, WDE, and HP1a, leads to ectopic expression of genes, many of which are normally expressed in testis. SETDB1 controls the accumulation of H3K9me3 over a subset of these genes without spreading into neighboring loci. At *phf7*, a regulator of male germ cell sexual fate, the H3K9me3 peak falls over the silenced testis-specific transcription start site. Furthermore, H3K9me3 recruitment to *phf7* and repression of testis-specific transcription is dependent on the female sex determination gene *Sxl*. Thus, female identity is secured by an H3K9me3 epigenetic pathway in which *Sxl* is the upstream female-specific regulator, SETDB1 is the required chromatin writer, and *phf7* is one of the critical SETDB1 target genes.

[1] Department of Genetics and Genome Sciences, Case Western Reserve University School of Medicine, Cleveland, OH 44106-4955, USA. Correspondence and requests for materials should be addressed to H.K.S. (email: hks@case.edu)

In metazoans, germ cell development begins early in embryogenesis when the primordial germ cells are specified as distinct from somatic cells. Specified primordial germ cells then migrate into the embryonic gonad, where they begin to exhibit sex-specific division rates and gene expression programs, ultimately leading to meiosis and differentiation into either eggs or sperm. Defects in sex-specific programming interferes with germ cell differentiation leading to infertility and germ cell tumors[1,2]. Successful reproduction, therefore, depends on the capacity of germ cells to maintain their sexual identity in the form of sex-specific regulation of gene expression.

In *Drosophila melanogaster*, germ cell sexual identity is specified in embryogenesis by the sex of the developing somatic gonad[3]. However, extrinsic control is lost after embryogenesis and sexual identity is preserved by a cell-intrinsic mechanism. The SEX-LETHAL (SXL) female-specific RNA binding protein is an integral component of the cell-intrinsic mechanism, as loss of SXL specifically in germ cells leads to a global upregulation of spermatogenesis genes and a germ cell tumor phenotype[4]. Remarkably, sex-inappropriate transcription of a single gene, *PHD finger protein 7 (phf7)*, a key regulator of male identity[5], is largely responsible for the tumor phenotype[4]. Depletion of *phf7* in mutants lacking germline SXL suppresses the tumor phenotype and restores oogenesis. Moreover, forcing PHF7 protein expression in ovarian germ cells is sufficient to disrupt female fate and give rise to a germ cell tumor. Interestingly, sex-specific regulation of *phf7* is achieved by a mechanism that relies primarily on alternative promoter choice and transcription start site (TSS) selection. Sex-specific transcription produces mRNA isoforms with different 5′ untranslated regions that affect translation efficiency, such that PHF7 protein is only detectable in the male germline[4–6]. Although the SXL protein is known to control expression post-transcriptionally in other contexts[7], the observation that germ cells lacking SXL protein show defects in *phf7* transcription argues that *Sxl* is likely to indirectly control *phf7* promoter choice. Thus, how this sex-specific gene expression program is stably maintained remains to be determined.

Here, we report our discovery that female germ cell fate is maintained by an epigenetic regulatory pathway in which SETDB1 (aka EGGLESS, KMT1E, and ESET) is the required chromatin writer and *phf7* is one of the critical SETDB1 target genes. SETDB1 trimethylates H3K9 (H3K9me3), a feature of heterochromatin[8,9]. Using tissue-specific knockdown approaches we establish that germ cell specific loss of SETDB1, its protein partner WINDEI [WDE, aka ATF7IP, MCAF1 and hAM[10]], and the H3K9me3 reader, HETEROCHROMATIN BINDING PROTEIN 1a [HP1a, encoded by the *Su(var)205* locus[11]], leads to ectopic expression of euchromatic protein-encoding genes, many of which are normally expressed only in testis. We further find that H3K9me3 repressive marks accumulate in a SETDB1 dependent manner at 21 of these ectopically expressed genes, including *phf7*. Remarkably, SETDB1 dependent H3K9me3 deposition is highly localized and does not spread into neighboring loci. Regional deposition is especially striking at the *phf7* locus, where H3K9me3 accumulation is restricted to the region surrounding the silent testis-specific TSS. Lastly, we find that H3K9me3 accumulation at many of these genes, including *phf7*, is dependent on *Sxl*. Collectively our findings support a model in which female fate is preserved by deposition of H3K9me3 repressive marks on key spermatogenesis genes.

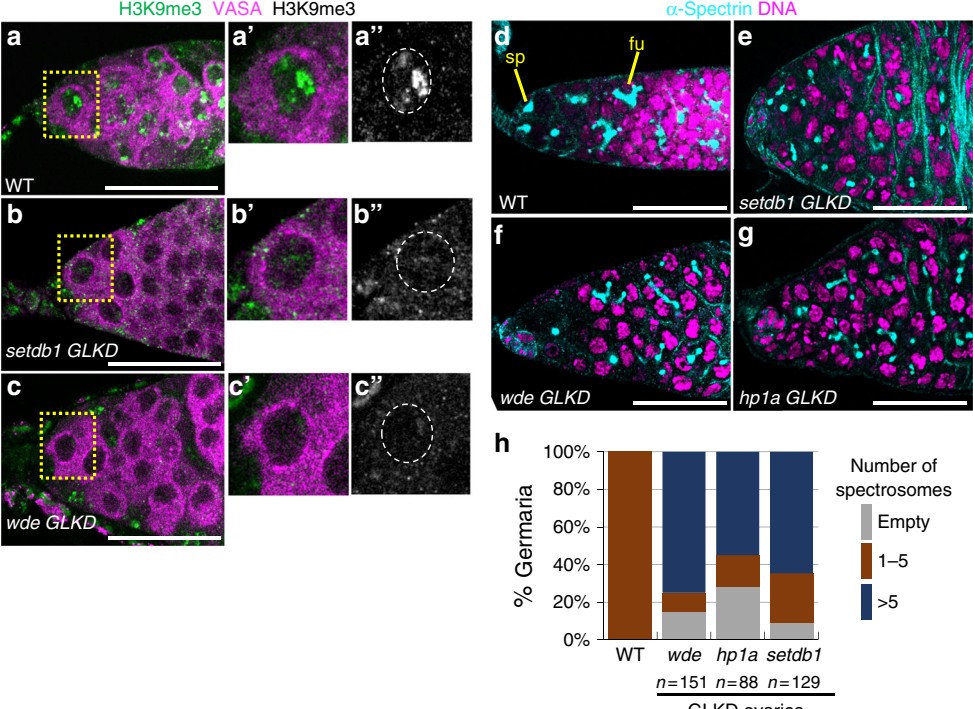

**Fig. 1** H3K9me3 is required for germ cell differentiation. **a–c** Reduced H3K9me3 staining in germ cells upon SETDB1 and WDE depletion (*GLKD*, see text for details*)*. Representative confocal images of a wild-type (WT), *setdb1 GLKD,* and *wde GLKD* germarium stained for H3K9me3 (green, white in **a″**–**c″**). Germ cells were identified by α-VASA staining (magenta). Scale bar, 25 μm. Insets show a higher magnification of a single germ cell, in which the nucleus is outlined by a white dashed line in **a″**–**c″**. **d–g** Undifferentiated germ cells accumulate in *setdb1 GLKD, wde GLKD*, and *hp1a GLKD* mutant germaria. Representative confocal images of wild-type and mutant germaria stained for DNA (magenta) and α-Spectrin (cyan) to visualize spectrosomes (sp), fusomes (fu), and somatic cell membranes. Scale bar, 25 μm. **h** Quantification of mutant germaria with 0, 1–5, and >5 round spectrosome-containing germ cells. The number of scored germaria (*n*) is indicated

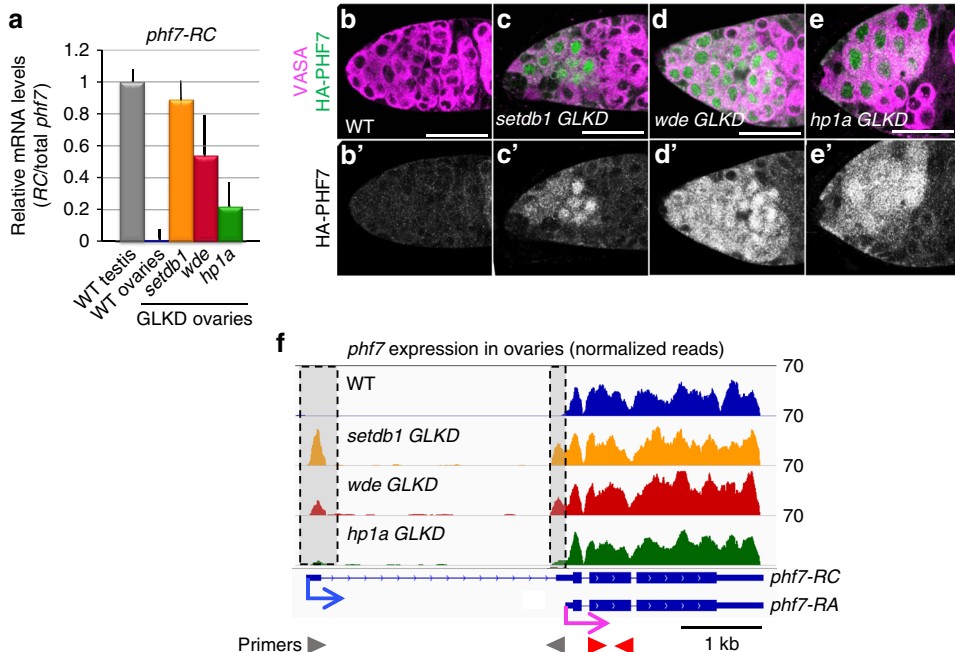

**Fig. 2** SETDB1, WDE, and HP1a depletion leads to female-to-male reprogramming at *phf7*. **a** Depletion of H3K9me3 pathway members leads to ectopic expression of the testis-specific *phf7-RC* isoform. RT-qPCR analysis of the testis *phf7-RC* transcript in wild-type testis, wild-type and mutant ovaries. Expression, normalized to the total level of *phf7*, is shown as fold change relative to testis. Primers are shown in panel F. Error bars indicate standard deviation (s.d.) of three biological replicates. **b–e** Depletion of H3K9me3 pathway members leads to ectopic expression of the testis-specific PHF7 protein. Ovaries from animals carrying an HA-PHF7 transgene stained for HA (green, white in B'-E'). Germ cells were identified by α-VASA staining (magenta). Scale bar, 25 μm. **f** RNA-seq data confirms ectopic expression of the testis-specific *phf7-RC* isoform. Genome browser view of the *phf7* locus. Tracks show RNA-seq reads aligned to the *Drosophila* genome (UCSC dm6). All tracks are viewed at the same scale. The screen shot is reversed so that the 5' end of the gene is on the left. The reads that are unique to the mutant ovaries are highlighted in gray. The two *phf7* transcripts, *phf7-RA* and *phf7-RC*, are indicated. *phf7-RC* is normally only expressed in testis (blue arrow). *phf7-RA* is normally expressed in ovaries (pink arrow). Primers for RT-qPCR are indicated by arrowheads: gray for *phf7-RC*, red for total *phf7*

## Results

**SETDB1, WDE and HP1a loss blocks germ cell differentiation.** Of the three *Drosophila* enzymes known to methylate H3K9, only SETDB1 is required for germline development[9]. Several studies reported that loss of SETDB1 caused a block in germ cell differentiation, characteristic of a germ cell tumor phenotype[12–15]. Because of the known connection between the germ cell tumor phenotype and ectopic testis gene transcription, we wondered whether SETDB1 played a role in silencing the expression of testis genes in female germ cells. Previous studies established that SETDB1 is important for Piwi-interacting small RNA (piRNA) biogenesis and transposable element (TE) silencing in germ cells[14,16,17]. However, mutations that specifically interfere with piRNA production, such as *rhino*, complete oogenesis[18–21]. Furthermore, our analysis of published RNA-sequencing (RNA-seq) data from *rhino* mutant ovaries[21] revealed only very minor effects on gene expression (Supplementary Fig. 1). Together these observations suggest that SETDB1 has a role in germ cell development that is unrelated to its canonical role in piRNA biogenesis and TE silencing.

We first confirmed that the loss of SETDB1 and its binding partner WDE specifically in germ cells was the cause of the germ cell tumor phenotype. To achieve Germ Line specific Knock Down (GLKD), we expressed an inducible RNA interference (RNAi) transgene with *nos-Gal4*, which is specifically expressed in germ cells. We demonstrated RNAi efficiency by showing that *setdb1 GLKD* and *wde GLKD* abolished the intense H3K9me3 staining foci observed in wild-type germ cells (Fig. 1a–c, Supplementary Fig. 2a–c, e). Furthermore, we found the oogenesis defects elicited by *setdb1* and *wde GLKD* to be similar, as judged by the number of round spectrosome like structures present in the germarium. The

spectrosome is a spherical α-Spectrin-containing structure that is normally found only in germline stem cells (GSCs) and its differentiating daughter cell, the cystoblast (~5 per germarium) (Fig. 1d). As differentiation proceeds, the round spectrosome elongates and branches out to form a fusome. We found that the majority of *setdb1* and *wde GLKD* mutant germaria contain 6 or more spectrosome containing germ cells (Fig. 1e, f, h). In wild-type, fusomes degenerate as the 16-cell germ cell cyst, consisting of an oocyte and 15 nurse cells, are enveloped by somatic follicle cells forming an egg chamber (Supplementary Fig. 2f). In mutants, however, we observed spectrosome containing germ cells enclosed by follicle cells (Supplementary Fig. 2g, h). This indicates that loss of SETDB1 and WDE in germ cells blocks differentiation, giving rise to a tumor phenotype.

Recently, a large-scale RNAi screen identified a role for H3K9me3 binding protein HP1a in germ cell differentiation[22]. In agreement with their findings, we observed that loss of HP1a in germ cells gave rise to germ cell tumors (Fig. 1g, h, Supplementary Fig. 2i). HP1a is required for gene silencing in other contexts[11]. This suggests that HP1a may act in a common pathway with SETDB1 and WDE in female germ cells.

**SETDB1, WDE and HP1a mutant ovaries express testis genes.** To investigate the possibility that the loss of H3K9me3 pathway members in female germ cells leads to ectopic testis gene expression, we first used RT-qPCR to assay for the presence of the testis-specific *phf7-RC* isoform in mutant ovaries. Using primer pairs capable of detecting *phf7-RC*, we found that *phf7-RC* is ectopically expressed in *setdb1*, *wde*, and *hp1a* mutant ovaries

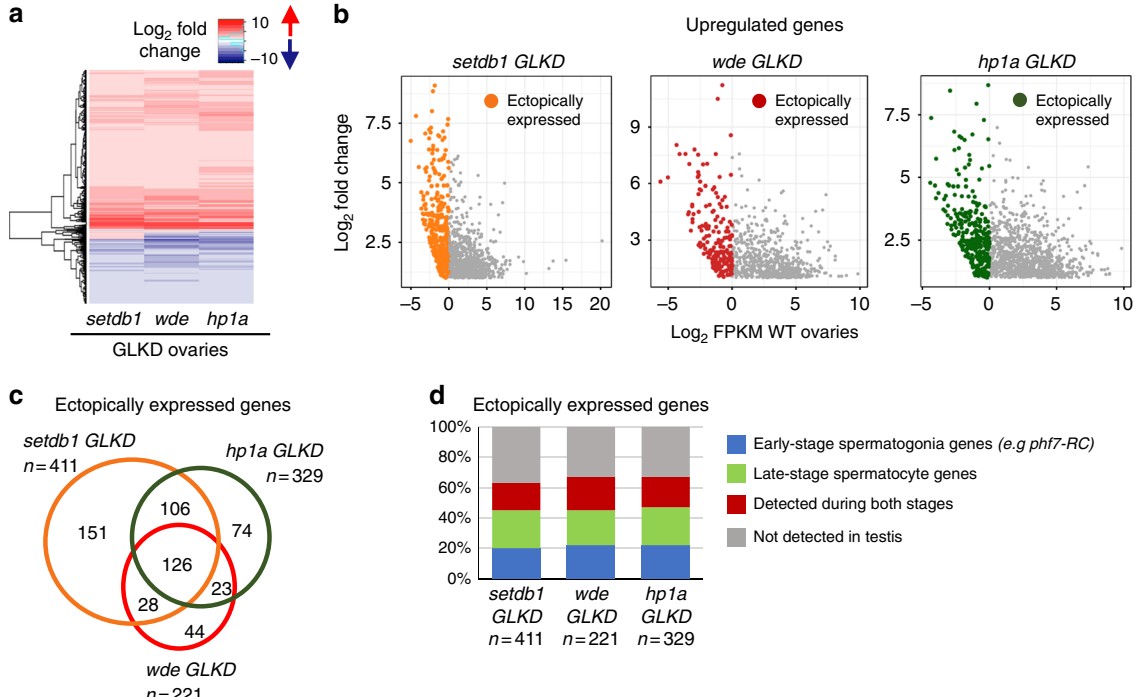

**Fig. 3** Depletion of H3K9me3 pathway members leads to ectopic expression of a similar gene set. **a** Comparisons of the deregulated gene expression profiles of *setdb1, wde* and *hp1a* GLKD ovaries reveals extensive similarities. Heat map comparing changes in gene expression in *setdb1, wde,* and *hp1a* GLKD ovaries compared to wild-type (WT) ovaries. Each row depicts a gene whose expression is deregulated at least 2-fold (FDR < 0.05) in all mutants when compared to wild-type. **b** Depletion of pathway members leads to ectopic expression of genes which are not normally expressed in ovaries. Scatter plots of significantly upregulated genes in *setdb1, wde,* and *hp1a* GLKD ovaries. The log$_2$ fold change in gene expression is plotted against the log$_2$ of the FPKM values in wild-type ovaries. Colored points indicate ectopically expressed genes (log$_2$ FPKM < 0 in WT ovaries). **c** Venn diagram showing overlap of ectopically expressed genes in *setdb1, wde,* and *hp1a* GLKD ovaries. The amount of overlap is significantly higher than expected stochastically ($P < 10^{-6}$). The significance of each two-way overlap was assessed using Fisher's exact test performed in R, in each case yielding $P < 10^{-6}$. The significance of the three-way overlap was assessed by a Monte Carlo simulation, yielding $P < 10^{-6}$. For each genotype, the number (n) of ectopically expressed genes is indicated. **d** A majority of ectopically expressed genes are normally expressed in testis. Bar chart showing the percentage of ectopically expressed genes in *setdb1, wde,* and *hp1a* GLKD ovaries that are normally expressed in wild-type testis. Genes are assigned into groups based on expression in wild-type and *bgcn* mutant testis (see text for details): early stage spermatogonia genes (>2-fold increase in *bgcn* testis compared to wild-type testis, in blue), late-stage spermatocyte genes (>2-fold decrease in *bgcn* testis compared to wild-type testis, in green) and genes detected during both stages (FPKM >1 in both samples, in red). Genes which are not expressed in either testis sample (FPKM < 1) are in gray. For each genotype, the number (n) of ectopically expressed genes is indicated

(Fig. 2a). Next, we asked whether ectopic *phf7-RC* expression correlates with ectopic PHF7 protein expression. Previous work using an HA-tagged *phf7* locus in the context of an 20 kb BAC rescue construct showed that PHF7 protein is normally restricted to testis[4,5]. We found that in contrast to wild-type ovaries, HA-PHF7 protein is detectable in the cytoplasm and in the nucleus of *setdb1, wde,* and *hp1a* mutant ovaries (Fig. 2b–e). We therefore conclude that the H3K9me3 pathway members are essential for suppression of testis-specific *phf7* transcription and PHF7 protein expression in female germ cells.

To gain a genome-wide view of the expression changes associated with the loss of H3K9me3 pathway members in germ cells, we used RNA-seq to compare the transcriptomes of *GLKD* mutant ovaries with wild-type ovaries from newborn (0–24 h) females. In agreement with our RT-qPCR analysis, we find that the testis-specific *phf7* transcript, *phf7-RC*, is ectopically expressed in *setdb1, wde,* and *hp1a* GLKD mutant ovaries (Fig. 2f). In addition to *phf7*, our differential analysis identified 1191 genes in *setdb1* GLKD mutant ovaries, 904 in *wde* GLKD ovaries, and 1352 in *hp1a* GLKD ovaries that are upregulated at least 2-fold relative to wild-type (FDR < 0.05; Supplementary Data 1–3). Additionally, 657 genes in *setdb1* GLKD mutant ovaries, 756 in *wde* GLKD ovaries, and 877 in *hp1a* GLKD ovaries are

downregulated (Supplementary Data 4–6). Comparison of the differential gene expression profiles of *setdb1* GLKD mutant ovaries with *wde* and *hpa1* GLKD mutant ovaries revealed extensive similarities, as expected for genes functioning in the same pathway (Fig. 3a).

Interestingly, all mutants express a set of upregulated genes that are normally not expressed in wild-type ovaries (FPKM < 1 in wild-type ovaries Fig. 3b, c, Supplementary Data 7–9). While we found that there was a significant overlap between the genes ectopically expressed in all three mutant backgrounds (Fig. 3c), we did not find that they were enriched for specific gene ontology terms. However, the pivotal role of *phf7* in controlling germ cell sex identity suggested to us that many of the ectopically expressed genes might be normally expressed in testis. To test this idea, we compared our data with published RNA-seq analysis of wild-type testis and *bgcn* mutant testis[23]. In spermatogenesis, *bgcn* is required for the undifferentiated spermatogonia to stop mitosis and transition into the spermatocyte stage. In *bgcn* mutants this transition is blocked and the testis are enriched for dividing spermatogonial cells. The comparison of the wild-type and mutant expression profiles can therefore be used to identify genes preferentially expressed in early-stage spermatogonia (>2-fold increase in *bgcn* compared to wild-type, in blue) and in late-stage

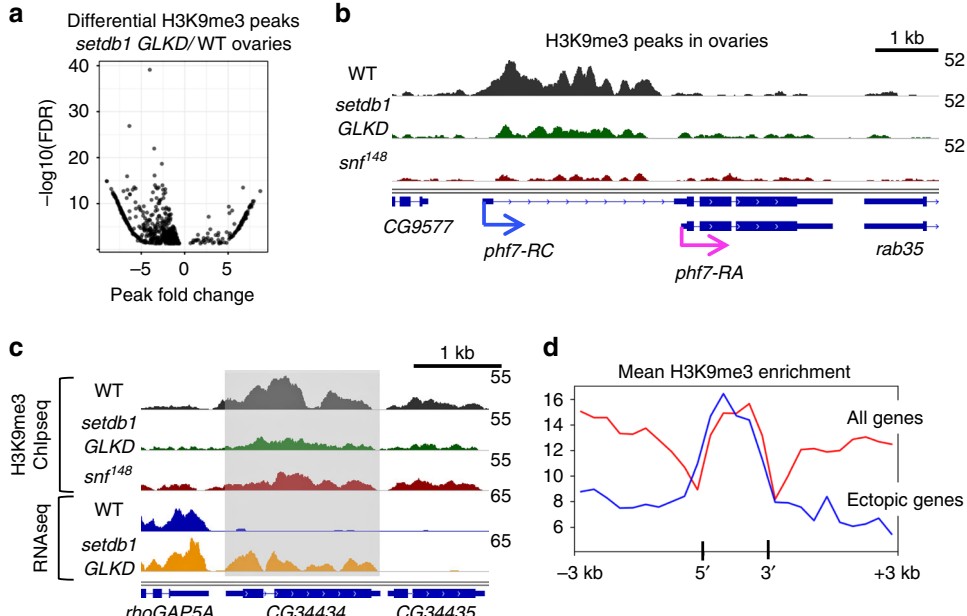

**Fig. 4** Loss of SETDB1 in female germ cells leads to H3K9me3 depletion on a select set of genes **a** Differential analysis of paired H3K9me3 ChIP-seq data sets identifies SETDB1-dependent H3K9me3 peaks. Scatter plot showing the significantly altered (FDR < 0.05) H3K9me3 peaks in *setdb1 GLKD* ovaries relative to wild-type (WT) ovaries. The x-axis is the $\log_2$ H3K9me3 enrichment of wild-type peaks subtracted from the $\log_2$ H3K9me3 enrichment of *setdb1 GLKD* peaks. Negative values indicate a reduction in H3K9me3 in mutant ovaries. **b** The H3K9me3 peak over the testis-specific *phf7-RC* TSS is decreased in *setdb1 GLKD* and *snf148* mutant ovaries. Genome browser view of *phf7* and neighboring genes *CG9577* and *rab35*. **c** In *setdb1 GLKD* ovaries, ectopic expression of *CG34434* is correlated with a decreased in H3K9me3 accumulation. H3K9me3 accumulation is also decreased in *snf148* mutant ovaries. Genome browser view of *CG34434* and its neighbors, *rhoGAP5A* and *CG34435*. RNA reads in wild-type (WT) and *setdb1 GLKD* are in blue and orange, respectively. The ChIP-seq reads are in black, green and red. *CG34434* is shaded. **d** Average enrichment profile indicates that the H3K9me3 peaks at the 21 SETDB1-regulated genes are localized over the gene body. The average H3K9me3 enrichment profile on the average gene body (transcription start site to transcription end site) scaled to 1500 base pairs, ±3 kb. In red, the average enrichment profile of euchromatic genes which display an H3K9me3 peak in wild-type ovaries. In blue, the average H3K9me3 profile of the 21 genes which are both ectopically expressed and display a loss of H3K9me3 enrichment in *setdb1 GLKD* ovaries

spermatocytes (>2-fold decrease in *bgcn* compared to wild type, in green). We also identified genes that are normally expressed in testis but are not differentially expressed (FPKM > 1 in both samples, in red) and genes that are not detectable in either sample (FPKM < 1, in gray). This analysis revealed that 63–67% of the ectopically expressed genes are expressed in testis (Fig. 3d; Supplementary Data 7–9). Because both early and late stage testis genes are ectopically expressed in the mutant ovaries, we conclude that the H3K9me3 pathway genes SETDB1, WDE and HP1a are required to repress spermatogenesis transcription in female germ cells.

This analysis also shows that 33–37% of the ectopically expressed genes are not normally expressed in gonads (gray, Fig. 3d), suggesting that the H3K9me3 pathway also represses somatic gene transcription. However, we did not identify a predominant tissue-specific signature amongst the remaining ectopically expressed genes (Supplementary Fig. 3). Furthermore, and despite ectopic expression of somatic genes, the mutant germ cells retained their germ cell identity, as evidenced by the presence of spectrosomes and fusomes, germ-cell specific organelles, as well as expression of the germ cell marker VASA (Fig. 1a–g). These results indicate that function of the H3K9me3 pathway in germ cells is not restricted to repressing the spermatogenesis gene expression program.

**H3K9me3 islands correlate with sex-specific gene expression.** Our studies raise the possibility that SETDB1 prevents linage-inappropriate gene transcription by mediating the deposition of H3K9me3 on its target loci. To test this idea directly, we performed H3K9me3 chromatin immunoprecipitation followed by sequencing (ChIP-seq) on wild-type and *setdb1 GLKD* ovaries. By limiting the differential peak analysis to euchromatin genes, we identified 746 H3K9me3 enrichment peaks in wild-type that were significantly altered in *setdb1 GLKD* ovaries (Fig. 4a). Whereas a majority of the gene associated peaks show the expected decrease in H3K9me3 enrichment (84%, 630/746), we also observed regions with an increase in H3K9me3 enrichment (15%, 116/746). How loss of SETDB1 might lead to an increase in H3K9me3 is not known, but the effect is most likely indirect.

At the *phf7* locus, H3K9me3 is concentrated over the silenced testis-specific TSS in wild-type ovaries (Fig. 4b). Reduction of this peak in *setdb1 GLKD* ovaries correlates with aberrant testis-specific transcription (Fig. 2f). These data suggest a functional link between the presence of repressive H3K9me3 chromatin and the silencing of *phf7* testis-specific isoform transcription.

We identified an additional 24 normally silenced euchromatic genes at which reduction of the H3K9me3 peak in *setdb1 GLKD* ovaries correlates with ectopic expression. Of these genes, 4 contain transposable element (TE) sequences. Prior studies have shown that H3K9me3 is enriched around euchromatic TE insertion sites, suggesting the possibility that transcriptional repression might result from spreading of H3K9me3 from a silenced TE[24–26]. The absence of TE sequences at *phf7* and the other 20 genes suggests H3K9me3 deposition is controlled by a different mechanism. Like *phf7*, the majority of these 20 SETDB1-regulated genes are normally expressed in spermatogenesis (Table 1). Furthermore, examination of their expression pattern in adult tissues, as reported in FlyAtlas[27], indicates that 8 of these

**Table 1 SETDB1/H3K9me3 regulated genes in ovaries**

| Gene | Expression pattern in adults (see text) | Predicted function |
|---|---|---|
| *Genes normally expressed only in testis, or genes with testis-specific transcripts* | | |
| *phf7* | Early stage testis specific transcript, *phf7-RC* | H3K4me2 binding |
| *skpE* | Early-stage spermatogonia | SKP1 gene family |
| CG12477 | Early-stage spermatogonia | Ring finger domain/E3 ligase |
| CG42299 | Both early and late | Zinc finger MIZ-type/E3 ligase |
| CG12061 | Both early and late | Sodium/calcium exchanger |
| CG13423 | Late-stage spermatocyte | Peptidase |
| CG15172 | Late-stage spermatocyte | Unknown |
| CG34434 | Late-stage spermatocyte | Unknown |
| CR43299 | Late-stage spermatocyte | ncRNA |
| *Genes normally expressed in testis and other tissues* | | |
| Lim1 | Early-stage spermatogonia; brain/CNS | Homeobox transcription factor |
| CG32506 | Early-stage spermatogonia; brain/CNS | Rab GTPase activating proteins |
| CG42613 | early-stage spermatogonia; brain/CNS | CUB domain |
| CG10440 | Late-stage spermatocyte; brain/CNS | BTB/POZ domain |
| CG10483 | Late-stage spermatocyte; brain/CNS | SPX and EXS domains |
| CG17636 | Late-stage spermatocyte; gut | Hydrolase |
| CG31202 | Late-stage spermatocyte; crop | Alpha-mannosidase class I |
| *Genes normally not expressed in testis* | | |
| CG12607 | No expression | Unknown |
| CG32679 | No expression | Secretory protein |
| CG15818 | Gut | Carbohydrate binding |
| MsR1 | Brain/CNS | Transmembrane G protein coupled receptor |
| Rab3-GEF | Brain/CNS | GDP-GTP exchange factor |

genes express at least one testis-specific isoform. Of the remaining genes, 7 express isoforms in the testis and other tissues, and 5 are not normally expressed in the adult testis. Further studies are needed to determine whether repression of these genes is as important for female germ cell development as *phf7*.

Previous studies have shown that ectopic PHF7 protein expression is sufficient to disrupt female fate and give rise to a germ cell tumor[4]. We therefore asked if ectopic PHF7 contributes to the *setdb1* GLKD mutant phenotype by generating double mutant females. We found that while loss of *phf7* did not restore oogenesis, there was a shift in the distribution of mutant phenotypes such that the majority of *phf7*[ΔN18]; *setdb1* GLKD double mutant ovarioles contained no germ cells (Supplementary Fig. 4). While these data indicate that *phf7* is a critical target of SETDB1 silencing, our finding that the phenotype was not fully rescued suggests that ectopic expression of one or more of the other SETDB1 target genes we identified in this study also contribute to the tumor phenotype.

Co-regulated genes are often clustered together in the genome. In *Drosophila*, about a third of the testis-specific genes are found in groups of three or more genes[28,29]. However, the 21 SETDB1-regulated genes we have identified do not fall within the previously identified testis gene clusters, nor are they clustered together in the genome (Fig. 5, Supplementary Table 1). Even on the X chromosome, where 11 of the 21 genes are located, the closest two genes are 100 kb away from each other. Thus, the SETDB1 regulated genes are not located within co-expression domains.

In fact, the H3K9me3 peaks at the 21 SETDB1-regulated genes are highly localized and do not spread into the neighboring genes (e.g. Fig. 4b, c, Supplementary Fig. 5). Averaging the H3K9me3 distribution over these 21 genes, scaled to 1.5 kb and aligned at their 5′ and 3′ ends, demonstrates a prominent enrichment over the gene body (Fig. 4d, in blue). In contrast, the average enrichment profile over all euchromatic genes with H3K9me3 peaks showed a broader pattern extending both upstream and

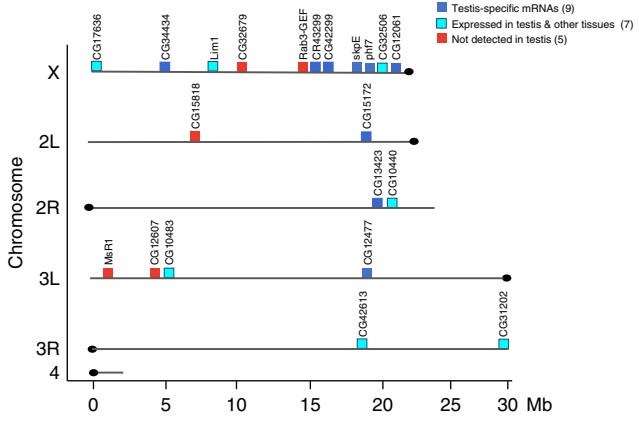

**Fig. 5** Distribution of the 21 SETDB1/H3K9me3 regulated genes in the genome. The 21 genes are not clustered together in the genome. Gene positions are shown on the six major chromosome arms. Chromosome length is indicated in megabase pairs (Mb). See Supplementary Table 1 for the exact position of each gene. The normal expression pattern of these genes is indicated as follows: genes with testis-specific isoform (dark blue), genes expressed in testis and other tissues (turquoise) and genes not normally expressed in adult testis (red). See Table 1 for details

downstream of the gene body (Fig. 4d, in red). Together these results clearly show that silencing involves formation of gene-specific blocks of H3K9me3 islands at a select set of testis genes.

**SXL loss in germ cells interferes with H3K9me3 accumulation.** Previous studies have established that sex-specific *phf7* transcription is controlled by the female sex determination gene *Sxl*[4]. Together with studies demonstrating that SXL protein is expressed in *setdb1* mutant germ cells[13], these data suggest that SXL acts upstream, or in parallel to SETDB1 to control *phf7* transcription. To assess the potential of a *Sxl*-mediated

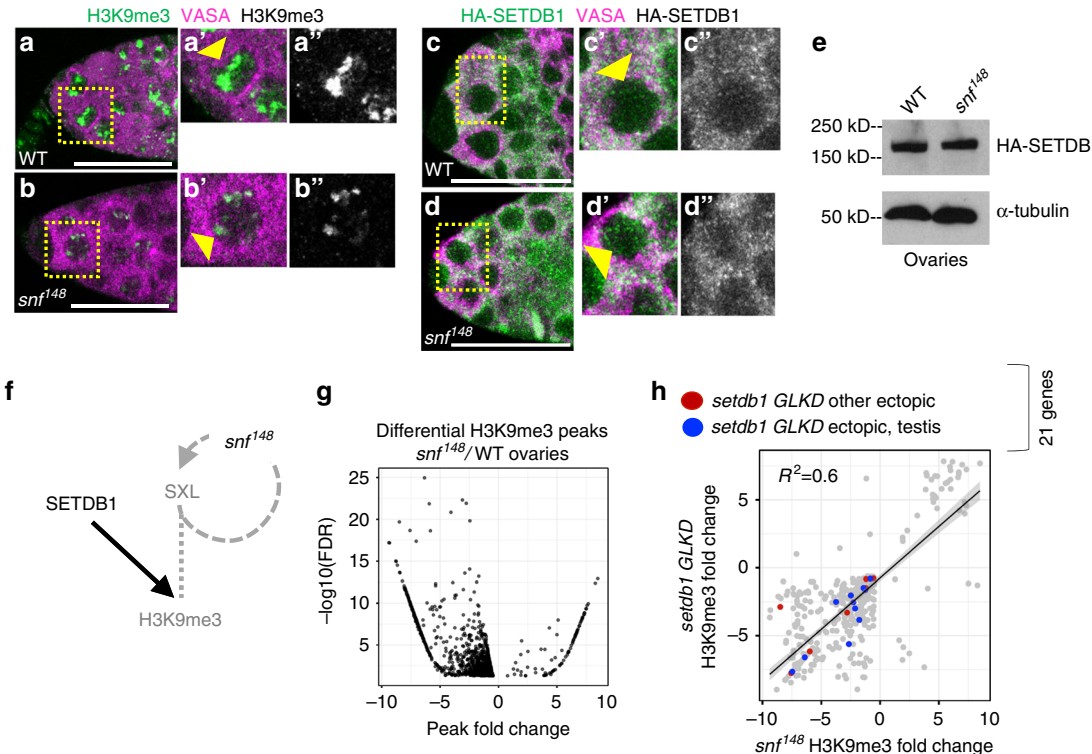

**Fig. 6** H3K9me3 distribution is affected in *snf[148]* mutant ovaries. **a, b** Reduced H3K9me3 staining in *snf[148]* mutant germ cells. Confocal images of a wild-type (WT) and *snf[148]* germaria stained for H3K9me3 (green, white in inset). Germ cells were identified by α-VASA staining (magenta). Scale bar, 25 μm. Insets show a higher magnification of a single germ cell, outlined by a dashed line. **c, d** SETDB1 protein localization is not altered in *snf[148]* mutant germ cells. Confocal images of a germaria from wild-type and *snf[148]* females carrying a copy an endogenously HA-tagged allele of *setdb1* stained for HA (green, white in inset). Germ cells were identified by α-VASA staining (magenta). Scale bar, 12.5 μm. Insets show a higher magnification of a single germ cell, outlined by a dashed line. **e** Western blot of ovarian lysates from wild-type and *snf[148]* females carrying a copy an endogenously HA-tagged allele of *setdb1* probed with an antibody against the HA tag. α-tubulin the loading control. **f** Diagram of genetic pathway controlling H3K9me3 accumulation in female germ cells. In WT, SXL collaborates with SETDB1 to regulate H3K9me3 accumulation. *snf[148]*, by virtue of the fact that it interferes with *Sxl* splicing, leads to germ cells without SXL protein. This in turn leads to a defect in H3K9me3 accumulation without interfering with SETDB1 protein accumulation. **g** Differential analysis of paired H3K9me3 ChIP-seq data sets identifies peak changes in *snf[148]* mutant ovaries. Scatter plot of significantly altered (FDR < 0.05) H3K9me3 peaks in *snf[148]* ovaries relative to wild-type (WT) ovaries. The x-axis is the $\log_2$ H3K9me3 enrichment of wild type peaks subtracted from the $\log_2$ H3K9me3 enrichment of *snf[148]* peaks. Negative values indicate a decreased H3K9me3 peak in mutant ovaries. **h** *setdb1 GLKD* and *snf[148]* influence H3K9me3 accumulation on a similar set of genes, including the genes ectopically expressed in *setdb1* GLKD ovaries. Plot comparing the significantly altered H3K9me3 peaks observed in *snf[148]* ovaries to *setdb1 GLKD* ovaries. Genes that are not normally expressed in ovaries but are ectopically expressed in *setdb1 GLKD* are labeled in red or in blue. Blue indicates genes which are normally expressed in testis

mechanism, we asked whether the loss of SXL in germ cells affects H3K9me3 accumulation. As with our earlier studies, we take advantage of the viable *sans-fille[148]* (*snf[148]*) allele to selectively eliminate SXL protein in germ cells without disrupting function in the surrounding somatic cells. SNF, a general splicing factor, is essential for *Sxl* autoregulatory splicing[7]. The viable *snf[148]* allele disrupts the *Sxl* autoregulatory splicing loop in female germ cells, leading to a failure in SXL protein accumulation, masculinization of the gene expression program (including *phf7*), and a germ cell tumor phenotype[4,30–32]. All aspects of the *snf[148]* mutant phenotype described to date are restored by germ cell-specific expression of a *Sxl* cDNA. Therefore, the analysis of *snf[148]* mutant ovaries directly informs us of *Sxl* function in germ cells. Interestingly, we found that that the intense H3K9me3 staining foci observed in wild-type germ cells was reduced in *snf[148]* mutants (Fig. 6a, b, Supplementary Fig. 2d, e). However, two lines of evidence indicate that SETDB1 protein expression, measured by an HA tag knocked into the endogenous locus[33], was not disrupted in *snf[148]* mutant germ cells. First, in whole mount immunostaining of both wild-type and *snf[148]* mutant germ cells, SETDB1 showed diffuse cytoplasmic staining and punctate

nuclear staining (Fig. 6c, d arrow head). Second, Western blot analysis of ovarian extracts showed that *snf[148]* tumors and wild-type ovaries have a similar level of SETDB1 protein (Fig. 6e, Supplementary Fig. 6). Our finding that H3K9me3 staining is disrupted, even though SETDB1 protein accumulation appears normal in *snf[148]* mutant ovaries, leads us to conclude that SXL and SETDB1 collaboratively promote H3K9me3-mediated silencing (Fig. 6f).

To directly test whether *Sxl* plays a role in controlling H3K9me3 deposition, we profiled the distribution of H3K9me3 by ChIP-seq in *snf[148]* mutant ovaries and compared it to the distribution in wild-type ovaries. By limiting the differential peak analysis to within 1 kb of euchromatic genes, we identified 1,039 enrichment peaks in wild-type that were significantly altered in *snf[148]* mutant ovaries, 91% of which show the expected decrease in H3K9me3 enrichment (Fig. 6g). When we compared the changes in *snf[148]* and *setdb1* GLKD mutants, we found a close correlation ($R^2 = 0.6$; Fig. 6h). The strong overlap between the regions displaying decreased H3K9me3 enrichment in *snf[148]* and *setdb1 GLKD* mutant ovaries suggests that SXL and SETDB1 influence H3K9me3 accumulation on the same set of genes

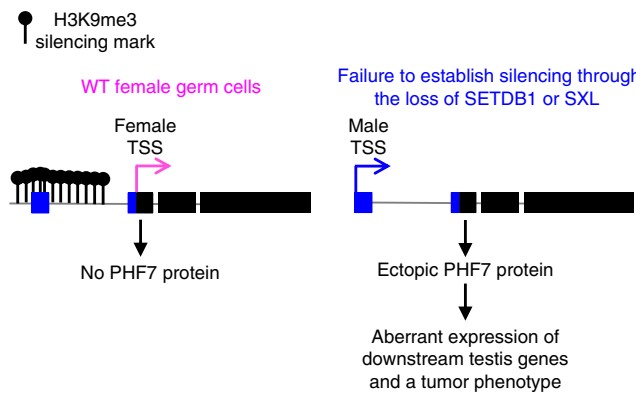

**Fig. 7** Schematic summary of discrete facultative heterochromatin island assembly at *phf7*. In female germ cells, SETDB1, together with SXL, directs assembly of a highly localized H3K9me3 domain around the testis-specific TSS. In germ cells lacking SETDB1 or SXL protein, the dissolution of the H3K9me3 domain correlates with ectopic testis-specific *phf7-RC* transcription and PHF7 protein expression. Ectopic PHF7 protein activity leads to activation of downstream testis genes and a tumor phenotype

(Fig. 6h), including *phf7* (Fig. 4b) and CG34434 (Fig. 4c). Based on these studies we conclude that SXL functions with SETDB1 in the assembly of H3K9me3 silencing islands in germ cells.

## Discussion

This study reveals a role for H3K9me3 chromatin, operationally defined as facultative heterochromatin, in securing female identity by silencing lineage-inappropriate transcription. We show that H3K9me3 pathway members, the H3K9 methyltransferase SETDB1, its binding partner WDE, and the H3K9 binding protein HP1a, are required for silencing testis gene transcription in female germ cells. Our studies further suggest a mechanism in which SETDB1, in conjunction with the female fate determinant SXL, controls transcription through deposition of highly localized H3K9me3 islands on a select subset of these genes. The male germ cell sexual identity gene *phf7* is one of the key downstream SETDB1 target genes. H3K9me3 deposition on the region surrounding the testis-specific TSS guaranties that no PHF7 protein is produced in female germ cells. In this model, failure to establish silencing leads to ectopic PHF7 protein expression, which in turn drives aberrant expression of testis genes and a tumor phenotype (Fig. 7).

Prior studies have established a role for SETDB1 in germline Piwi-interacting small RNA (piRNA) biogenesis and TE silencing[14,16,17]. However, piRNAs are unlikely to contribute to sexual identity maintenance as mutations that specifically interfere with piRNA production, such as *rhino*, do not cause defects in germ cell differentiation[18–21] or lead to global changes in gene expression (Supplementary Fig. 1). These findings, together with our observation that *rhino* does not control sex-specific *phf7* transcription, suggests that the means by which SETDB1 methylates chromatin at testis genes is likely to be mechanistically different from what has been described for piRNA-guided H3K9me3 deposition on TEs.

We find that H3K9me3 accumulation at many of these genes, including *phf7*, is dependent on the presence of SXL protein. Thus, our studies suggest that SXL is required for female-specific SETDB1 function. SXL encodes an RNA binding protein known to regulate its target genes at the posttranscriptional levels[7]. SXL control may therefore be indirect. However, studies in mammalian cells suggest that proteins with RNA binding motifs are important for H3K9me3 repression[34,35], raising the tantalizing possibility that SXL might play a more direct role in governing

testis gene silencing. Further studies will be necessary to clarify how the sex determination pathway feeds into the heterochromatin pathway.

*phf7* stands out among the cohort of genes regulated by facultative heterochromatin because of its pivotal role in controlling germ cell sexual identity[4,5]. Because ectopic protein expression is sufficient to disrupt female fate, tight control of *phf7* expression is essential. *phf7* regulation is complex, employing a mechanism that includes alternative promoter usage and TSS selection. We report here that SETDB1/H3K9me3 plays a critical role in controlling *phf7* transcription. In female germ cells, H3K9me3 accumulation is restricted to the region surrounding the silent testis-specific transcription start site. Dissolution of the H3K9me3 marks via loss of SXL or SETDB1 protein is correlated with transcription from the upstream testis-specific site and ectopic protein expression, demonstrating the functional importance of this histone modification. Together, these studies suggest that maintaining the testis *phf7* promoter region in an inaccessible state is integral to securing female germ cell fate.

Although the loss of H3K9me3 pathway members in female germ cells leads to the ectopic, lineage-inappropriate transcription of hundreds of genes, our integrative analysis identified only 21 SETDB1/H3K9me3 regulated genes. Given that one of these genes is *phf7* and that ectopic PHF7 is sufficient to destabilize female fate[4,5], it is likely that inappropriate activation of a substantial number of testis genes is a direct consequence of ectopic PHF7 protein expression. How PHF7 is able to promote testis gene transcription is not yet clear. PHF7 is a PHD-finger protein that preferentially binds to H3K4me2[5], a mark associated with poised, but inactive genes and linked to epigenetic memory[36–38]. Thus, one simple model is that ectopic PHF7 binds to H3K4me2 marked testis genes to tag them for transcriptional activation.

It will be interesting to explore whether any of the other 20 SETDB1/H3K9me3 regulated genes also have reprogramming activity. In fact, ectopic fate-changing activity has already been described for the homeobox transcription factor Lim1 in the eye-antenna imaginal disc[39]. However, whether Lim1 has a similar function in germ cells is not yet known. Intriguingly, protein prediction programs identify three of the uncharacterized testis-specific genes as E3 ligases ([http://www.gene2f.org])[40]. SkpE is a member of the SKP1 gene family, which are components of the Skp1-Cullin-F-box type ubiquitin ligase. CG12477 is a RING finger domain protein, most of which are believed to have ubiquitin E3 ligases activity. CG42299 is closely related to the human small ubiquitin-like modifier (SUMO) E3 ligase NSMCE2. Given studies that have linked E3 ligases to the regulation of chromatin remodeling[41,42], it is tempting to speculate that ectopic expression of one or more of these E3 ligases will be sufficient to alter cell fate. Future studies focused on this diverse group of SETDB1/H3K9me3 regulated genes and their role in reprogramming may reveal the multiple layers of regulation required to secure cell fate.

The SETDB1-mediated mechanism for maintaining sexual identity we have uncovered may not be restricted to germ cells. Recent studies have established that the preservation of sexual identity is essential in the adult somatic gut and gonadal cells for tissue homeostasis[43–47]. Furthermore, the finding that loss of HP1a in adult neurons leads to masculinization of the neural circuitry and male specific behaviors[48] suggests a connection between female identity maintenance and H3K9me3 chromatin. Thus, we speculate that SETDB1 is more broadly involved in maintaining female identity.

Our studies highlight an emerging role for H3K9me3 chromatin in cell fate maintenance[49]. In the fission yeast *S. pombe*, discrete facultative heterochromatin islands assemble at meiotic genes that are maintained in a silent state during vegetative growth[50,51]. Although less well understood, examples in

mammalian cells indicate a role for SETDB1 in lineage-specific gene silencing[52–56]. Thus, silencing by SETDB1 controlled H3K9 methylation may be a widespread strategy for securing cell fate. Interestingly, H3K9me3 chromatin impedes the reprogramming of somatic cells into pluripotent stem cells (iPSCs). Conversion efficiency is improved by depletion of SETDB1[57–59]. If erasure of H3K9me3 via depletion of SETDB1 alters the sexually dimorphic gene expression profile in reprogrammed cells, as it does in *Drosophila* germ cells, the resulting gene expression differences may cause stem cell dysfunction, limiting their therapeutic utility.

## Methods

**Drosophila stocks and culture conditions**. Fly strains were kept on standard medium at 25 °C unless otherwise noted. Knockdown studies were carried out with the following lines generated by the *Drosophila* Transgenic RNAi Project[60]: *setdb1*-P{TRiP.HMS00112} (BDSC #34803, RRID: BDSC_34803), *Su(var)205/HP1a*-P{TRiP.GL00531} (BDSC #36792, RRID: BDSC_36792), and *wde*-P{TRiP.HMS00205} (BDSC #33339, RRID: BDSC_33339). Different conditions were used to maximize the penetrance of the germ cell tumor phenotype. For knockdown of *setdb1* and *HP1a* the *nos-Gal4;bam-Gal80* driver was used[61], crosses were set up at 29 °C and adults were aged 3–5 days prior to gonad dissection. For *wde* knockdown the *nos-Gal4* driver was used (BDSC #4937, RRID: BDSC_4937)[62], crosses were set up at 18 °C and adults were transferred to 29 °C for 2 days prior to gonad dissection. The following *Drosophila* stocks were also used in this study: *snf*[148] (BDSC #7398, RRID: BDSC_7398)[30], *HA-setdb1*[33], *phf7*[ΔN18] and *PBac{3XHA-Phf7}*[5]. Wild-type ovaries are either sibling controls or *y*[1] *w*[1] (BDSC #1495, RRID: BDSC_1495).

For each experiment described below, sample sizes were not predetermined by statistical calculations, but were based on the standard of the field. In a pool of control or experimental animals, specimens of the correct age and genotype were selected randomly and independently from different vials/bottles. Data acquisition and analysis were not performed blindly.

**Immunofluorescence and image analysis**. *Drosophila* gonads were fixed and stained according to standard procedures with the following primary antibodies: mouse α-Spectrin (1:100, Developmental Studies Hybridoma Bank [DSHB] 3A9 RRID: AB_528473], rabbit α-H3K9me3 (1:1,000, Active Motif Cat# 39162, RRID: AB_2532132], rat α-HA (1:500, Roche Cat# 11867423001, RRID: AB_390919), mouse α-Sxl (1:100, DSHB M18 RRID: AB_528464), rabbit α-Vasa (1:2,000, a gift from the Rangan lab), and rat α-Vasa (1:100, DSHB RRID: AB_760351). Staining was detected by FITC (1:200, Jackson ImmnoResearch Labs) or Alexa Fluor 555 (1:200, Life Technologies) conjugated species appropriate secondary antibodies. TO-PRO-3 Iodide (Fisher, Cat# T3605) was used to stain DNA. Images were taken on a Leica SP8 confocal with 1024 × 1024 pixel dimensions, a scan speed of 600 Hz, and a frame average of 3. Sequential scanning was done for each channel and three Z-stacks were combined for each image. Processed images were compiled with Gnu Image Manipulation Program (GIMP) and Microsoft PowerPoint. The GIMP Hue-saturation tool was used to assign appropriate colors to merged panels. All staining experiments were replicated at least two times. The "*n*" in the figure legends represents the number of germaria scored from a single staining experiment.

Fluorescence intensity quantification of H3K9me3 was measured in 5 wild-type and mutant germaria using GIMP and normalized to fluorescence intensity of DNA to control for the number of cells. Each image was obtained under identical conditions and consisted of three Z-stacks, with the germline stem cells in the plane of focus.

**Western analysis**. Ovary extracts for Western blots were prepared from hand-dissected ovaries from 100 females homogenized in 2X sample buffer (100 mM TRIS, pH 6.8, 10% β−Mercaptoethanol, 4% SDS, 20% glycerol, 0.1% Bromophenol Blue). Westerns were performed according to standard procedures with the following primary antibodies: rat α-HA high affinity (1:500, Roche # 11867423001, RRID: AB_390919), mouse α alpha-tubulin (1:500, DSHB #AA4.3, RRID: AB_579793), Enhanced chemiluminescence (ECL) was used for detection, with the following secondary antibodies: ECL goat anti-rat IgG HRP (1:2,000, Fisher # 45-001-200, RRID: AB_772207), and ECL sheep anti-mouse IgG HRP (1:2,000, Fisher # 45-000-679, RRID: AB_772210). For Western Blot Quantitation: Three replicates were used and the relative densities for each band were calculated with ImageJ [https://imagej.nih.gov/ij/]. Adjusted densities were determined by dividing the H3K9me3 relative densities by that of their corresponding loading controls. Uncropped scans are presented in Supplementary Fig. 6.

**qRT-PCR and data analysis**. RNA was extracted from dissected ovaries using TRIzol (Invitrogen, Cat# 15596026) and DNase (Promega, Cat# M6101). Quantity and quality were measured using a NanoDrop spectrophotometer. cDNA was generated by reverse transcription using the SuperScript First-Strand Synthesis System Kit (Invitrogen, Cat# 11904018) using random hexamers. Quantatative real-time PCR was performed using Power SYBR Green PCR Master Mix

(ThermoFisher, Cat# 4367659) with the Applied Biosystems 7300 Real Time PCR system. PCR steps were as follows: 95 °C for 10 min followed by 40 cycles of 95 °C for 15 s and 60 °C for 1 min. Melt curves were generated with the following parameters: 95 °C for 15 s, 60 °C for 1 min, 95 °C for 15 s, and 60 °C for 15 s. Measurements were taken in biological triplicate with two technical replicates. The *phf7-RC* levels were normalized to the total amount *phf7*. Relative transcript amount were calculated using the 2-ΔΔCt method[63]. Primer sequences for measuring the total *phf7* and *phf7-RC* levels were: for total *phf7*, forward GAGCT-GATCTTCGGCACTGT and reverse GCTTCGATGTCCTCCTTGAG; for *phf7-RC* forward AGTTCGGGAATTCAACGCTT and reverse GAGATAGCCCTGCAGCCA.

**RNA-seq and data analysis**. For wild-type, *setdb1 GLKD, wde GLKD*, and *HP1a GLKD* ovaries: Total RNA was extracted from dissected ovaries using standard TRIzol (Invitrogen, Cat# 15596026) methods. RNA quality was assessed with Qubit and Agilent Bioanalyzer. Libraries were generated using the Illumina TruSeq Stranded Total RNA kit (Cat# 20020599). Sequencing was completed on 2 biological replicates of each genotype with the Illumina HiSeq 2500 v2 with 100 bp paired end reads. Sequencing reads were aligned to the *Drosophila* genome (UCSC dm6) using TopHat (2.1.0)[64]. Differential analysis was completed using CuffDiff (2.2.1)[65]. Genes were considering differentially expressed if they exhibited a two-fold or higher change relative to wild-type with a False Discovery Rate (FDR) < 0.05. Heat maps were generated using the heatmap.2 function of the gplots R package. Scatter plots were generated using ggplot function in R.

Genes that were expressed in mutant (FPKM ≥ 1) but not expressed in wild-type ovaries (FPKM < 1) were called ectopic. Genes normally expressed in testes were identified by interrogating the published mRNA-seq data sets GSE86974[23] [http://www.ncbi.nlm.nih.gov/geo/], as above. Genes whose expression levels were two-fold or higher in *bgcn* mutant testis relative to wild-type testis were called "early-stage testis genes". Genes whose expression levels were two-fold or higher in wild-type testis relative to *bgcn* mutant testis were called "late-stage testis genes". Genes that were not differentially expressed but expressed in testis (FPKM ≥ 1) were called simply "testis genes". The RNA-seq data on Fly Atlas[27] was used to identify genes with testis-specific isoforms, *i.e.* no expression in any other adult tissue.

Tissue specific clustering of the ectopically expressed genes not normally expressed in testis was performed to identify tissue-specific signatures. Expression values normalized to the whole fly were extracted from FlyAtlas. Heatmaps to compare the tissue expression profile of these genes per tissue were generated in R with heatmap.2 (gplots). Genes were clustered and normalized per row.

Screen shots are from Integrated Genome Viewer (IGV). To account for the differences in sequencing depth when creating IGV screenshots, the processed RNAseq alignment files were scaled to the number of reads in the wild-type file. This was done with Deeptools bigwigCompare using the scale Factors parameter with a bin size of 5.

For *rhino* mutant ovaries, differentially expressed genes were identified utilizing published mRNA-seq datasets available from the National Center for Biotechnology Information's GEO database [http://www.ncbi.nlm.nih.gov/geo/] under accession number GSE55824[21] as described above.

**ChIP-seq and data analysis**. For each chromatin immunoprecipitation, 400 pairs of *Drosophila* ovaries were dissected in PBS plus protease inhibitors. Tissues were fixed in 1.8% formaldehyde for 10 min at room temperature and quenched with 225 mM glycine for 5 min. Tissues were washed twice and stored at −80 °C for downstream applications. Samples were lysed using the protocol from Lee et al., 2006. Tissue was placed in lysis buffer 1 (140 mM HEPES pH 7.5, 200 mM NaCl, 1 mM EDTA, 10% glycerol, 0.5% NP-40, 0.25% Triton X-100), homogenized using sterile beads, and rocked at 4 °C for 10 min. Tissue was then washed 10 min at 4 °C in lysis buffer 2 (10 mM Tris pH 8, 200 mM NaCl, 1 mM EDTA, 0.5 mM EGTA). Tissues were then placed in 1.5 mL lysis buffer 3 (10 mM Tris pH 8, 100 mM NaCl, 1 mM EDTA, 0.5 mM EGTA, 0.1% Na-deoxycholate, 0.5% N-lauroylsarcosine). All buffers were supplemented with protease inhibitors. Chromatin was sheared to 200–700 base pairs using the QSONICA sonicator (Q800R). The chromatin lysate was incubated overnight at 4 °C with H3K9me3 antibodies pre-bound to magnetic beads. The beads were prepared as follows: 25 μl Protein A and 25 μl Protein G Dynabeads (Invitrogen, Cat# 10002D and 10004D) per sample were washed twice with ChIP blocking buffer (0.5% Tween 20, 5 mg/mL BSA), then blocked by rocking at 4 °C for 1 h in ChIP blocking buffer, and then conjugated to 5 μg H3K9me3 antibody (Abcam Cat# 8898 RRID: AB_306848) by rocking in new ChIP blocking buffer at 4 °C for 1 h. Following immunoprecipitation, the samples were washed 6 times with ChIP-RIPA buffer (10 mM Tris-HCl pH 8, 1 mM EDTA, 140 mM NaCl, 1% Triton X-100, 0.1% SDS, 0.1% Na-Deoxycholate), 2 times with ChIP-RIPA/500 buffer (ChIP-RIPA + 500 mM NaCl), 2 times with ChIP-LiCl buffer (10 mM Tris-HCl pH 8, 1 mM EDTA, 250 mM LiCl, 0.5% NP-40, 0.5% Na-deoxycholate), and twice with TE buffer. DNA was eluted from beads with 50 μl elution buffer (10 mM Tris-HCl pH 8, 5 mM EDTA, 300 mM NaCl, 0.1% SDS) and reverse crosslinked at 65 °C for 6 h. Beads were spun down and eluted DNA was transferred to a new tube and extracted using phenol-chloroform extraction. All buffers were supplemented with protease inhibitors.

ChIP sequencing libraries were prepared using the Rubicon Genomics Library Prep Kit (Cat# R440406) with 16 amplification cycles. DNA was cleaned and

assessed for quality with Qubit and Agilent Bioanalyzer. Sequencing was completed on 2 biological replicates of each genotype with the Illumina HiSeq 2500 v2 with 50 bp single end reads.

H3K9me3 reads were aligned to the *Drosophila* genome (UCSC dm6) using bowtie2 (2.2.6)[66], and duplicate reads were removed with samtools (1.3)[67]. Peaks were called with MACS (2.1.20150731) using the broadpeaks option with all other paramaters set to default[68]. Differential peak analysis on all replicates was completed with the DIFFBIND program (2.4.8), using summits = 500 and the DESeq2 package [http://bioconductor.org/packages/release/bioc/vignettes/DiffBind/inst/doc/DiffBind.pdf]. The mean peak concentration (log$_2$) was calculated by normalizing reads to the total library size and subtracting the corresponding input reads. Differential peak fold changes were calculated by subtracting wild-type mean concentrations from mutant mean concentrations. Mutant peaks were considered significantly altered relative to wild-type if they had a False Discovery Rate (FDR) < 5%. The average H3K9me3 deposition on genes in wild-type ovaries was generated with deeptools (2.5.3), using normalized ChIP reads from wild-type ovaries[69]. Screen shots are from Integrated Genome Viewer (IGV).

## Data availability

The data that support the findings of this study are available from the corresponding author upon reasonable request. RNA-seq and ChIP-seq data sets generated during the course of this study are available from the National Center for Biotechnology Information's GEO database under accession number GSE109852. RNAseq datasets from testis[23] and *rhino* mutant ovaries[21] are available from the National Center for Biotechnology Information's GEO database under accession numbers GSE86974 and GSE55824. Tissue enrichment datasets are available from the FlyAtlas and FlyAtlas.2.

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

## Acknowledgements

We thank Jean-Rene Huynh, Mark Van Doren, Prash Rangan, the Bloomington *Drosophila* Stock Center and the Iowa Developmental Studies Hybridoma Bank for fly stocks and antibodies; Alex Miron, Neil Molyneaux, Ricky Chan, Ulrich Ness, Olivia Corradin, Dan Factor, Alina Saiakhova for help with the bioinformatic analysis; Heather Broihier, Michelle Longworth, Ron Conlon, Peter Harte, and Peter Scacheri for helpful suggestions, discussions, and comments on the manuscript; and Jane Heatwole for fly food. This work was supported by the National Institutes of Health, R01GM102141 and R01GM129478 to H.K.S. and T32GM008056 to A.E.S. Imaging was performed using equipment purchased through NIH S10OD016164.

## Author contributions

Conceptualization: A.E.S and H.K.S.; Investigation, methodology, and analysis: A.E.S and L.S.K.; Manuscript writing, reviewing, and editing: A.E.S and H.K.S.; Supervision: H.K.S.

## Additional information

**Competing interests:** The authors declare no competing interests.

