## [Peer Review File · Nature Communications]

Reviewer #1 (Remarks to the Author):

Germ cells need to maintain the correct sexual identity for gametogenesis. Failures to do so causes infertility and can lead to germ cell tumors. Additionally, maintenance of sexual identity may be needed in tissues outside the gonads. Importantly, early mechanisms used to identify germ cell sex occurs early and needs to be maintained throughout development to allow correct patterns of transcription. Mechanisms controlling maintenance are not fully understood.

Smolko and colleagues show that the female germline identity is preserved through an epigenetic mechanism involving the readers and writers of H3K9me3. These studies combine genetic and immunohistochemical analyses with high throughput RNA sequencing and chromatin immunoprecipitation to demonstrate the role of the key silencing factors in repression of a set of key spermatogenesis genes. Critically, they show that SETDB1 is required for H3K9me3 deposition at an alternative transcription start site in the gene *phf7*, which has a central role in promoting the male germ cell sexual identity. Interestingly, H3K9me3 is focal and shows little spreading. Deposition of H3K9me3 requires to *Sex-lethal*, the critical female identity gene, providing a complete pathway towards maintenance of the female pathway. Smolko and colleagues note that SETDB1 has additional roles in the germline, including piRNA repression, but this pathway is not involved in maintenance of sexual identity.

This is an interesting and important study that advances our understanding of a fundamental regulatory cascade. In addition, these studies open up several new lines of investigation. The authors note that although loss of function of SETDB1, HP1 and WDE individually cause misregulation of over a thousand genes, the overlap of all three genotypes reveals the core 21 genes, of which one is *phf7*. The authors suggest that these differences might account from mis-regulation of *phf7*, yet this suggestion does not seem to be reasonable, considering of the 21 is *phf7*. It seems equally likely that it is differences in genetic background responses, since only one genotype was examined. Regardless, it will be interesting to understand how *phf7* promotes the male program in the background of a female differentiation plan, as well as examining other factors in the overlap set.

Minor criticisms:

1. Fig. 1 describes the SETDB1/ HP1/ WDE knockdown phenotype-but it is hard to know what the complete phenotype is-the authors need to show the whole ovary-are the tumors similar to those found in the absence of the differentiation factor, BAM?
2. The authors need to define what is meant by testis genes and testis-specific (are these genes never expressed outside the testis?)
3. Do the authors have any insight into why 11 of 21 genes are on the X chromosomes? This seems surprising.
4. In the investigation of the *Sex lethal* role in the repression pathway, the authors test *snf* mutants. They note that SETDB1 accumulation appears normal, but the immunohistochemical analysis is hard to see- a western should be included.
5. The authors do not discuss 6F- what is shown here?

Typos or clarification:

1. Pg. 3, line 12- this instead of the
2. Pg. 3, line 16, what is meant by these?
3. More complete figure legends, including the definition of abbreviations, such as GLKD.
4. Table 1: the organization is hard to understand-perhaps it should be organized in alphabetical order.
5. Fig. 4: the figure legend states CG34435 but the figure says CG34434.

Reviewer #2 (Remarks to the Author):

This work uses a combination of genetic experiments and genomics with bioinformatics to understand the role of H3K9 methyltransferase SETDB1 in female identity in *Drosophila*. The authors find that germ cell-specific loss of the H3K9me3 pathway genes causes the inappropriate expression of testis genes. They also show that SETDB1 is required for H3K9me3 accumulation on a select subset of the silenced testis genes. They further show that repression of the testis-specific transcript of one of the silenced testis genes, *phf7*, is dependent on the female sex determination gene *Sxl*. This paper suggests that female fate may be preserved by deposition of H3K9me3 marks on key spermatogenesis genes.

Overall, the paper has some relevant information that will be useful for people working on germ line development in *Drosophila*. However, comments below indicate aspects that need significant revision before publication. This revision is important to determine the actual impact of the work.

Major comments:

1. The authors shall consider performing RNAi knock-down of *phf7* and other testis-specific genes identified in their experiments in *setdb1* GLKD and examining whether the germ cell tumor phenotype observed in *setdb1* GLKD is ameliorated.

2. Related to 1, a number of genes are derepressed in *setdb1* GLKD. Figure 3D shows that not only testis genes but also many other genes are derepressed in *setdb1* GLKD. These ectopically expressed genes in *setdb1* GLKD may also contain neuron-specific genes, for example. To corroborate their model of female fate being preserved by deposition of H3K9me3 marks on key spermatogenesis genes, it is important to show that repression of key spermatogenesis genes is critical for female fate in germ line development.

3. The authors claim that "this function of SETDB1 is unrelated to its canonical role in piRNA biogenesis and silencing of transposable elements". However, this paper also lacks data to support this claim. It is known that SETDB1 is involved in transposon silencing mediated by the Piwi-piRNA pathway. Thus, a number of transposons should be derepressed in *setdb1* GLKD. Have the authors examined levels of transposon transcripts in *setdb1* GLKD?

4. It is very likely that mutations in *snf*, a general splicing factor, affect splicing of *Sxl* but also many other genes, which could lead to misexpression of many genes. Thus, it is conceivable that the observed changes of H3K9me3 accumulation (on *phf7*) may have nothing to do with *SXL* and SETDB1.

Reviewer #3 (Remarks to the Author):

General Comments:

This interesting and well documented study beautifully lays out an elegant mechanism whereby function of the H3K9 histone methyltransferase *Setdb1* and its partners act in female early germ cells to repress expression of a key regulator of male early germ cell state. The study should be of wide interest to the germ cell, sex determination, and chromatin fields.

Specific Comments:

Figure 2 and legend: The label in Panel A is a bit misleading, as it could be interpreted that all

the phf7 mRNA expressed in testes comes from the upstream promoter (in other words the downstream promoter is off in testes). Yet, careful reading of the legend indicates that the gray column in figure 2A was set to 1. Which is it? The two have different implications: is the phenomenon a switch to different promoter in the two sexes or just addition of usage of the upstream promoter in testes, while the downstream promoter still fires? Panel B and its legend should include information that the RNA-Seq data come from ovaries. The legend states that reads unique to the mutants are highlighted by gray shading, but gray shading is hardly apparent in the figure as printed. There are wording errors in the sentence that begins "Beneath is the Ref Seq...."

Figure 3 and legend: How are the genes ordered in panels A and B? If by clustering – please show cluster diagram at left. Are the genes in the same order in A and B or in different order?

I note that in D, ~ 40% of the ectopically expressed genes were not detected in testis RNA. Is this because

H3K9me3 in ovaries represses many genes that would be on in brain or gut or other tissues? How much of these gray, not expressed in testis genes are under control of phf7.

Figure 4 and legend: The label for Panel B should be SetdbGLKD / WT Ovaries. The term VS. does not give us information about what -5 compared to +5 means.

Figure 6 and legend: In the legend for panel C, "regulated" should be "regulate". Label to Panel D should be snf148 /WT instead of "VS", similar as for E. The last sentence is confusing.

Page 4, lines 24-26: Isn't NOS expressed again in growing oocytes? Is the NOS-Gal 4 also so expressed? If so, then the statement that NOS-Gal 4 drives expression "largely restricted to pre-meiotic germ cells" would not be correct, at least for ovaries.

Throughout: All the results should be stated in the past tense. Some of examples of corrections needed: page 5 line 6; page 5 lines 10 -11; page 5 line 19; page 6 line 8, etc..

Page 5, line 21: Fix wording: should read "...the male-specific transcript...". Clarify with writing and in figure label and legend whether both the male and female transcripts are produced in the mutant ovaries, or does the male transcript turn on and the female transcript turn off, as implied by the label of Figure 2 panel A.

Page 5, lines 25 -26: I thought ovarioles in newly eclosed (unfed) females had egg chambers up to stage 7 (onset of yolk deposition) – thus they contain many cell types not found in germ line tumors. If so, the statement here seems misleading.

Page 6, line 9: The testis specific phf7 transcript is only slightly apparent in the RNA Seq reads from hp1a KD ovaries shown in Figure 2B, so this statement should be modified.

Page 6 bottom and figure 3D: Are the "testis genes" shown in red in panel 3D expressed in testis and elsewhere or are they expressed much higher in testis than in other tissues? Actually, the genes are in all scells, so the wording should be changed here to refer to the transcripts or to expression.

Page 7 top and figure 3: A good 3rd to 40% of the genes ectopically up regulated in ovaries when Setdb1 or its partners have been knocked down in germ cells are not normally detected as expressed in testis, indicating that Setdb1 and H3K9me3 regulate much more in ovaries than the male germ cell specific program. This begs the question of how much of the phenotype observed in ovaries is due to the mis-expression of male germ cell specific programs. What is the effect if the male specific transcript of Phf7 is knocked down in or deleted at the same time as knock down of Setdb1 or its partners?

How much of the effect of loss of SETDB1 is due to mis-expression of phf7? It would be interesting to examine the double loss of function SETDB1; phf7 ^{-/-}. Will the testis specific transcripts now no longer fire, but the non-testis transcripts still fire (gray in Figure 3D)?

Page 7, lines 12 -15, and elsewhere: The authors should fill the readers in on whether there are other enzymes in Drosophila that can make the H3K9me3 chromatin mark, and what role these may play.

Page 7, line 17 - 22: The enrichment by ChIP for of H3K9me3 over the phf7 – RC promoter region shown in figure 4B is not lost – but reduced. The authors should change the wording here accordingly.

Figure 6: It would be useful to show additional panels with black and white images of the HA – SetDB1 signal alone in figure 6B as well, so that the readers can more easily see the cytoplasmic distributions without interference from the Vasa signal.

Reviewer #4 (Remarks to the Author):

Preserving sexual identity is required for gametogenesis. To do execute this during Drosophila oogenesis select testis-enriched genes are repressed. Previously, the authors have show that ectopic expression of a male specific isoform of phf7 (phf7-RC) in the female germ line leads to tumors of undifferentiated cells. In this manuscript the authors propose that SETDB1, and pathway members, silence testis genes (such as phf7-RC) via H3K9me3 deposition. The authors should address the following questions/suggestions prior to publication.

- o In Figure 1A-1C Smolko et al. are trying to show that germ line loss of setdb1 and wde leads to ablation of H3K9me3 mark. But, as the channels are merged it makes it challenging to see this reduction. To make this clearer, please quantitate the levels of H3K9me3 and separate the channels.

- o In Figure 2 the authors quantitate levels of phf7-RC in WT testis, WT ovaries, setdb1, wde, and hp1a GLKD, showing an increase in levels of phf7-RC upon loss of SETDB1, WDE, and HP1a. This finding would be more meaningful if PHF7 protein levels were also increased in setdb1, wde, and hp1a GLKD compared to undifferentiated cells in WT ovaries.

- o In Figure 3C the authors show that there are several overlapping targets of setdb1, wde, and hp1a GLKD. Do these genes have an enriched-GO term associated with them?

- o In Figure 6A-B Smolko et al. show that SXL also regulates expression of testis genes during oogenesis. To better show that H3K4me3 levels are reduced and SETDB1 levels are unaffected, please quantitate histone mark and HA levels.

We thank the four reviewers for investing the time necessary to provide detailed and thoughtful suggestions. In aggregate, these comments and questions have substantially improved this manuscript. Our responses, in **red**, are given in a point-by-point manner below. We have uploaded a revised version of the manuscript in which the major changes to the text are in **red**.

In response to the reviewer's collective input and the manuscript checklist, the revised manuscript now includes the following major additions:

- 1) A new abstract that is less than 150 words.
- 2) Images in Fig. 1e-g that were recolored to avoid confusion for color-blind readers.
- 3) Quantitation of H3K9me3 staining in Supplementary Fig. 2e, and the addition of black and white images to better illustrate H3K9me3 staining in Fig. 1a-c, Fig. 6 a,b and Supplementary Fig. 2a-d.
- 4) New images of the entire wild-type and mutant ovarioles are provided in Supplementary Fig. 2f-i.
- 5) New data indicating that SETDB1 has a role in germ cell development that is unrelated to its canonical role in piRNA biogenesis is presented in Supplementary Fig. 1.
- 6) New data showing that PHF7 protein is detectable in mutant ovarioles is presented in Fig. 2b-e
- 7) Image in Fig. 3a was replaced to show the cluster diagram, and Image in Fig. 3b was replaced to better convey our point that a significant fraction of genes upregulated in mutant ovaries were not normally expressed in ovaries.
- 8) New Supplementary Fig. 3, showing that the non-gonadal genes derepressed in H3K9me3 pathway mutants are not characteristic of a single tissue lineage.
- 9) New double mutant analysis presented in Supplementary Fig. 4 showing that ectopic PHF7 contributes to the *setdb1 GLKD* tumor phenotype.
- 10) New Western blots (and quantitation) showing that *snr¹⁴⁸* tumors and wild-type ovaries have a similar level of SETDB1 protein Fig. 6e and Supplementary Fig. 6. Black and white images were added to Fig. 6 c,d to better illustrate HA-SETDB1 staining in mutant germ cells.

Reviewer #1 (Remarks to the Author):

Germ cells need to maintain the correct sexual identity for gametogenesis. Failures to do so causes infertility and can lead to germ cell tumors. Additionally, maintenance of sexual identity may be needed in tissues outside the gonads. Importantly, early mechanisms used to identify germ cell sex occurs early and needs to be maintained throughout development to allow correct patterns of transcription. Mechanisms controlling maintenance are not fully understood.

Smolko and colleagues show that the female germline identity is preserved through an epigenetic mechanism involving the readers and writers of H3K9me3. These studies combine genetic and immunohistochemical analyses with high throughput RNA sequencing and chromatin immunoprecipitation to demonstrate the role of the key silencing factors in repression of a set of key spermatogenesis genes. Critically, they show that SETDB1 is required for H3K9me3 deposition at an alternative transcription start site in the gene *phf7*, which has a central role in promoting the male germ cell sexual identity. Interestingly, H3K9me3 is focal and shows little spreading. Deposition of H3K9me3 requires to *Sex-lethal*, the critical female identity gene, providing a complete pathway towards maintenance of the female pathway. Smolko and colleagues note that SETDB1 has additional roles in the germline, including piRNA repression, but this pathway is not involved in maintenance of sexual identity.

This is an interesting and important study that advances our understanding of a fundamental regulatory cascade. In addition, these studies open up several new lines of investigation. The authors note that although loss of function of SETDB1, HP1 and WDE individually cause misregulation of over a thousand genes, the overlap of all three genotypes reveals the core 21 genes, of which one is *phf7*. The authors suggest that these differences might account from mis-regulation of *phf7*, yet this suggestion does not seem to be reasonable, considering of the 21 is *phf7*. It seems equally likely that it is differences in genetic background responses, since only one genotype was examined. Regardless, it will be interesting to understand how *phf7* promotes the male program in the background of a female differentiation plan, as well as examining other factors in the overlap set.

Thank you. In regards to future studies, we share this reviewers concern that while *phf7* stands out among the cohort of SETDB1-regulated genes because of its known pivotal role in controlling germ cell sexual identity (Shapiro-Kulnane et al., 2015; Yang et al., 2012), it is possible that one or more of the other 20 SETDB1-regulated genes may also have reprogramming activity. Indeed, double mutant studies requested by Reviewer #2 and #3 (Supplementary Fig. 4) support this conclusion. The text at the end of this section (lines 180-182) now reads: *“While these data indicate that *phf7* is a critical target of SETDB1 silencing, our finding that the phenotype was not fully rescued suggests that ectopic expression of one or more of the other SETDB1 target genes we identified in this study also contribute to the tumor phenotype.”*

Minor criticisms:

1. Fig. 1 describes the SETDB1/ HP1/ WDE knockdown phenotype-but it is hard to know what the complete phenotype is-the authors need to show the whole ovary-are the tumors similar to those found in the absence of the differentiation factor, BAM?

We have now provided images of the entire ovariole are provided in Supplementary Fig. 2f-i. We describe our observations in the text (lines 90-93) as follows: *“In wild-type, fusomes degenerate as the 16-cell germ cell cyst, consisting of an oocyte and 15 nurse cells, are enveloped by somatic follicle cells forming an egg chamber (Supplementary Fig. 2f). In mutants, however, we observed spectrosome containing germ cells enclosed by follicle cells (Supplementary Fig. 2g,h). This indicates that loss of SETDB1 and WDE in germ cells blocks differentiation, giving rise to a tumor phenotype.”* In the next paragraph, we document that HP1a *GLKD* gives rise to a similar phenotype.

We note that we (Fig. 1h) and others (Clough et al., 2007; Clough et al., 2014; Rangan et al., 2011; Wang et al., 2011; Yan et al., 2014) have shown that the tumor phenotype is more variable than what is found in the absence of *bam*.

2. The authors need to define what is meant by testis genes and testis-specific (are these genes never expressed outside the testis?)

We apologize for not providing enough information regarding definitions in our initial submission. To clarify our meaning, we have revised the RNA-seq analysis section (lines 124-129), beginning with: *“While we found that there was a significant overlap between the genes ectopically expressed in all three mutant backgrounds (Fig. 3c), we did not find that they were enriched for specific gene ontology terms. However, the pivotal role of *phf7* in controlling germ cell sex identity suggested to us that many of the ectopically expressed genes might be*

*normally expressed in testis. To test this idea, we compared our data with published RNA-seq analysis of wild-type testis and *bgcn* mutant testis (Shan et al., 2017)."*

Only after the ChIP analysis, which focused our attention on the core 21 genes, do we differentiate between those genes that are normally *only* expressed in testis and those genes which are normally expressed in testis as well as in other adult tissues. The text (lines 168-174) now reads: "*Like *phf7*, the majority of these 20 SETDB1-regulated genes are normally expressed in spermatogenesis (Table 1). Furthermore, examination of their expression pattern in adult tissues, as reported in FlyAtlas (Leader et al., 2018), indicates that 8 of these genes express at least one testis-specific isoform. Of the remaining genes, 7 express isoforms in the testis and other tissues, and 5 are not normally expressed in the adult testis.*" The text in Table 1 has also been revised.

3. Do the authors have any insight into why 11 of 21 genes are on the X chromosomes? This seems surprising.

We agree that this result is surprising, especially in light of the conventional wisdom that most genes with high levels of testis expression when compared to ovaries, are underrepresented on the X chromosome. However, we feel that any discussion of this point is too speculative to be incorporated into the revised manuscript.

4. In the investigation of the Sex lethal role in the repression pathway, the authors test *snf* mutants. They note that SETDB1 accumulation appears normal, but the immunohistochemical analysis is hard to see- a western should be included.

This is an excellent suggestion. A Western blot is now included (lines 214-215) "*Western blot analysis of ovarian extracts showed that *snf*^{f48} tumors and wild-type ovaries have a similar level of SETDB1 protein (Fig. 6e, Supplementary Fig. 6).*"

5. The authors do not discuss 6F- what is shown here?

We apologize for this omission. The section describing this figure, now Fig. 6h, now reads (lines 219-224): "*To directly test whether *Sxl* plays a role in controlling H3K9me3 deposition, we profiled the distribution of H3K9me3 by ChIP-seq in *snf*^{f48} mutant ovaries and compared it to the distribution in wild-type ovaries. By limiting the differential peak analysis to within 1 kb of euchromatic genes, we identified 1,039 enrichment peaks in wild-type that were significantly altered in *snf*^{f48} mutant ovaries, 91% of which show the expected decrease in H3K9me3 enrichment (Fig. 6g). When we compared the changes in *snf*^{f48} and *setdb1* GLKD mutants, we found a close correlation ($R^2=0.6$; Fig. 6h).*"

Typos or clarification:

1. Pg. 3, line 12- this instead of the

Fixed

2. Pg. 3, line 16, what is meant by these?

Fixed

3. More complete figure legends, including the definition of abbreviations, such as GLKD.

Fixed

4. Table 1: the organization is hard to understand-perhaps it should be organized in alphabetical order.

We have organized the genes into three groups based on their normal expression pattern. We have added the following subheadings to clarify this point. “*genes normally expressed only in testis, or genes with testis-specific transcripts*”; “*genes normally expressed in testis and other tissues*”; “*genes normally not expressed in testis*”.

5. Fig. 4: the figure legend states CG34435 but the figure says CG34434.

Done. Thank you for noticing.

Reviewer #2 (Remarks to the Author):

This work uses a combination of genetic experiments and genomics with bioinformatics to understand the role of H3K9 methyltransferase SETDB1 in female identity in *Drosophila*. The authors find that germ cell-specific loss of the H3K9me3 pathway genes causes the inappropriate expression of testis genes. They also show that SETDB1 is required for H3K9me3 accumulation on a select subset of the silenced testis genes. They further show that repression of the testis-specific transcript of one of the silenced testis genes, *phf7*, is dependent on the female sex determination gene *Sxl*. This paper suggests that female fate may be preserved by deposition of H3K9me3 marks on key spermatogenesis genes.

Overall, the paper has some relevant information that will be useful for people working on germ line development in *Drosophila*. However, comments below indicate aspects that need significant revision before publication. This revision is important to determine the actual impact of the work.

We have made every attempt to fully address all of your comments in the revised manuscript. We believe that these revisions have resulted in a significantly improved manuscript.

Major comments:

1. The authors shall consider performing RNAi knock-down of *phf7* and other testis-specific genes identified in their experiments in *setdb1* GLKD and examining whether the germ cell tumor phenotype observed in *setdb1* GLKD is ameliorated.

This is an excellent suggestion. We were only able to make the *phf7^{null}*; *setdb1* GLKD double mutants, as reagents for the other 20 genes are not currently available. The data are reported in Supplementary Fig. 4, and discussed on (lines 175-182). The text reads: “*Previous studies have shown that ectopic PHF7 protein expression is sufficient to disrupt female fate and give rise to a germ cell tumor*(Shapiro-Kulnane et al., 2015). *We therefore asked if ectopic PHF7 contributes to the setdb1 GLKD mutant phenotype by generating double mutant females. We found that while loss of phf7 did not restore oogenesis, there was a shift in the distribution of mutant phenotypes such that the majority of phf7^{ΔN18}; setdb1 GLKD double mutant ovarioles contained no germ cells* (Supplementary Fig. 4). *While these data indicate that phf7 is a critical target of SETDB1 silencing, our finding that the phenotype was not fully rescued suggests that ectopic expression of one or more of the other SETDB1 target genes we identified in this study also contribute to the tumor phenotype.*”

We were not surprised that the double mutant females were not fertile because SETDB1 clearly does more than regulating *phf7* transcription. But, we did see that the phenotype was nudged towards a less tumorous phenotype indicating that *phf7* plays a role in tumorigenesis.

2. Related to 1, a number of genes are derepressed in *setdb1* GLKD. Figure 3D shows that not only testis genes but also many other genes are derepressed in *setdb1* GLKD. These ectopically expressed genes in *setdb1* GLKD may also contain neuron-specific genes, for

example. To corroborate their model of female fate being preserved by deposition of H3K9me3 marks on key spermatogenesis genes, it is important to show that repression of key spermatogenesis genes is critical for female fate in germ line development.

This is a good point. Hundreds of genes are ectopically expressed in mutant germ cells, 63%-67% of which are normally expressed in testis. Thus, it is fair to conclude that the H3K9me3 pathway also represses other lineage-inappropriate genes. We now incorporate this point in the text as follows (lines 141-148). *“This analysis also shows that 33%-37% of the ectopically expressed genes are not normally expressed in gonads (gray, Fig. 3d), suggesting that the H3K9me3 pathway also represses somatic gene transcription. However, we did not identify a predominant tissue-specific signature amongst the remaining ectopically expressed genes (Supplementary Fig. 3). Furthermore, and despite ectopic expression of somatic genes, the mutant germ cells retained their germ cell identity, as evidenced by the presence of spectrosomes and fusomes, germ-cell specific organelles, as well as expression of the germ cell marker VASA (Fig. 1 a-g). These results indicate that function of the H3K9me3 pathway in germ cells is not restricted to repressing the spermatogenesis gene expression program.”*

We note, however, that despite expression of somatic genes, these mutant germ cells retain their germ cell identity, as evidenced by the presence of spectrosome like structures and VASA protein expression (Fig. 1). Retention of germ cell identity, together with ectopic expression of many genes normally expressed in testis, including *phf7*, leads to the conclusion that one function of SETDB1/H3K9me3 is maintenance of female identity via control of *phf7* transcription. Direct support is provided by the results of the new double mutant analysis (Supplementary Fig. 4), requested by this reviewer, and by previously published work showing that forcing PHF7 in wild-type germ cells is sufficient to promote a female to male switch in gene expression and a tumor phenotype (Shapiro-Kulnane et al., 2015; Yang et al., 2012). In addition to *phf7* we have identified 20 other genes whose expression is likely to be directly controlled by H3K9me3. While we do not have the reagents in hand to study the importance of controlling expression of the other 20 genes in female germ cells, we speculate a bit about their importance in the discussion section (lines 276-287).

3. The authors claim that “this function of SETDB1 is unrelated to its canonical role in piRNA biogenesis and silencing of transposable elements”. However, this paper also lacks data to support this claim. It is known that SETDB1 is involved in transposon silencing mediated by the Piwi-piRNA pathway. Thus, a number of transposons should be derepressed in *setdb1* GLKD. Have the authors examined levels of transposon transcripts in *setdb1* GLKD?

Thank you for bringing this oversight to our attention. Previous studies have already reported that SETDB1 is important for transposon silencing (Rangan et al., 2011; Sienski et al., 2015; Yu et al., 2015). We have added data in Supplementary Fig. 1 showing that *rhino* mutations, which specifically interfere with germline piRNA production (Klattenhoff et al., 2009; Mohn et al., 2014; Volpe et al., 2001; Zhang et al., 2014), does not disrupt sex-specific *phf7* transcriptional regulation or lead to global changes in gene expression. Moreover, loss of *rhino* does not disrupt oogenesis. Together these findings indicate we have uncovered a role for SETDB1 that is unrelated to its canonical role in piRNA biogenesis and TE silencing.

These data are presented in Supplementary Fig. 1. The text now reads (lines 72-78): *“Previous studies established that SETDB1 is important for Piwi-interacting small RNA (piRNA) biogenesis and transposable element (TE) silencing in germ cells (Rangan et al., 2011; Sienski et al., 2015; Yu et al., 2015). However, mutations that specifically interfere with piRNA production, such as rhino, complete oogenesis (Klattenhoff et al., 2009; Mohn et al., 2014; Volpe et al., 2001; Zhang*

et al., 2014). Furthermore, our analysis of published RNA-sequencing (RNA-seq) data from rhino mutant ovaries (Mohn et al., 2014) revealed only very minor effects on gene expression (Supplementary Fig. 1). Together these observations suggest that SETDB1 has a role in germ cell development that is unrelated to its canonical role in piRNA biogenesis and TE silencing.”

And the discussion now reads (lines 241-248): *“Prior studies have established a role for SETDB1 in germline Piwi-interacting small RNA (piRNA) biogenesis and TE silencing (Rangan et al., 2011; Sienski et al., 2015; Yu et al., 2015). However, piRNAs are unlikely to contribute to sexual identity maintenance as mutations that specifically interfere with piRNA production, such as rhino, do not cause defects in germ cell differentiation (Klattenhoff et al., 2009; Mohn et al., 2014; Volpe et al., 2001; Zhang et al., 2014) or lead to global changes in gene expression (Supplementary Fig. 1). These findings, together with our observation that rhino does not control sex-specific phf7 transcription, suggests that the means by which SETDB1 methylates chromatin at testis genes is likely to be mechanistically different from what has been described for piRNA-guided H3K9me3 deposition on TEs.”* As this is clearly a hypothesis, we have removed this statement from the abstract.

4. It is very likely that mutations in *snf*, a general splicing factor, affect splicing of *Sxl* but also many other genes, which could lead to misexpression of many genes. Thus, it is conceivable that the observed changes of H3K9me3 accumulation (on *phf7*) may have nothing to do with *SXL* and *SETDB1*.

We apologize for not providing enough background information about the viable but sterile *snf*¹⁴⁸ allele in our initial submission. We have revised the text (lines 204-208) to provide this additional information: *“The viable *snf*¹⁴⁸ allele disrupts the *Sxl* autoregulatory splicing loop in female germ cells, leading to a failure in *SXL* protein accumulation, masculinization of the gene expression program (including *phf7*) and a germ cell tumor phenotype (Chau et al., 2009; Chau et al., 2012; Nagengast et al., 2003; Shapiro-Kulnane et al., 2015). All aspects of the *snf*¹⁴⁸ mutant phenotype described to date are restored by germ cell-specific expression of a *Sxl* cDNA (Chau et al., 2009; Chau et al., 2012; Nagengast et al., 2003; Shapiro-Kulnane et al., 2015). Therefore, the analysis of *snf*¹⁴⁸ mutant ovaries directly informs us of *Sxl* function in germ cells.”* We are therefore confident that the H3K9me3 defect we observe in *snf*¹⁴⁸ mutants is due to the loss of *SXL* protein in germ cells.

Reviewer #3 (Remarks to the Author):

General Comments:

This interesting and well documented study beautifully lays out an elegant mechanism whereby function of the H3K9 histone methyltransferase *Setdb1* and its partners act in female early germ cells to repress expression of a key regulator of male early germ cell state. The study should be of wide interest to the germ cell, sex determination, and chromatin fields.

Thank you.

Specific Comments:

Figure 2 and legend: The label in Panel A is a bit misleading, as it could be interpreted that all the *phf7* mRNA expressed in testes comes from the upstream promoter (in other words the downstream promoter is off in testes). Yet, careful reading of the legend indicates that the gray column in figure 2A was set to 1. Which is it? The two have different implications: is the

phenomenon a switch to different promoter in the two sexes or just addition of usage of the upstream promoter in testes, while the downstream promoter still fires?

This is a good point that has led us to more carefully explain transcription at the *phf7* locus both within the text and in the legend to Fig. 2. Briefly, only the longer *phf7* transcript is normally expressed in testis (*phf7-RC*). The other shorter transcript *phf7-RA* is normally expressed in ovaries. Because *phf7-RA* overlaps entirely with *phf7-RC* it is not entirely clear whether *phf7-RA* is expressed in testis. However, as illustrated below, when we compare the amount of *phf7-RC* (normalized to rp49) to total *phf7* (normalized to rp49) in testis by RT-qPCR, *phf7-RC* appears to be the majority transcript.

To avoid confusion, we have modified the text to read (lines 103-105): “we first used RT-qPCR to assay for the presence of the testis-specific *phf7-RC* isoform in mutant ovaries. Using primer pairs capable of detecting *phf7-RC*, we found that *phf7-RC* is ectopically expressed in *setdb1*, *wde*, and *hp1a* mutant ovaries (Fig. 2a).”

Panel B and its legend should include information that the RNA-Seq data come from ovaries. The legend states that reads unique to the mutants are highlighted by gray shading, but gray shading is hardly apparent in the figure as printed. There are wording errors in the sentence that begins “Beneath is the Ref Seq....”

We have added information to Fig. 2, and the legend to emphasize that the RNA-seq data is from ovaries (now Fig 2f). Dotted lines now surround the shaded areas to increase visibility, and the wording errors have been fixed.

Figure 3 and legend: How are the genes ordered in panels A and B? If by clustering – please show cluster diagram at left. Are the genes in the same order in A and B or in different order?

We apologize for this oversight. In Fig. 3a each row depicts a gene whose express is deregulated at least 2-fold (FDR<0.05) in all mutants when compared to wild-type. The rows are ordered by clustering and the diagram is now shown on the left.

This comment made us think more deeply about how best to convey our main point in Fig. 3b, which was to illustrate that a significant fraction of the upregulated genes in mutant ovaries were ectopic, *i.e.* not normally expressed in wild-type ovaries. We therefore replaced the heat map with three different scatter plots in which the \log_2 fold change in gene expression is plotted against the \log_2 FPKM in wild-type ovaries. Colored points indicate ectopically expressed genes (\log_2 FPKM < 0 in WT ovaries).

I note that in D, ~ 40% of the ectopically expressed genes were not detected in testis RNA. Is this because H3K9me3 in ovaries represses many genes that would be on in brain or gut or other tissues?

This question led us to examine the expression of these genes in other adult tissues. These new data are presented in Supplementary Fig. 3. We found that while expression of these genes is seen in other tissues such as the brain and gut, no consistent tissue-specific signature was identified.

We elaborate on this finding in the text (lines 141-148): *“This analysis also shows that 33%-37% of the ectopically expressed genes are not normally expressed in gonads (gray, Fig. 3d), suggesting that the H3K9me3 pathway also represses somatic gene transcription. However, we did not identify a predominant tissue-specific signature amongst the remaining ectopically expressed genes (Supplementary Fig. 3). Furthermore, and despite ectopic expression of somatic genes, the mutant germ cells retained their germ cell identity, as evidenced by the presence of spectrosomes and fusomes, germ-cell specific organelles, as well as expression of the germ cell marker VASA (Fig. 1 a-g). These results indicate that function of the H3K9me3 pathway in germ cells is not restricted to repressing the spermatogenesis gene expression program.”*

How much of these gray, not expressed in testis genes are under control of phf7.

This is an interesting question. The answer to this question would require a set of new studies focused on the consequences of ectopic PHF7 protein expression. The answer may be yes—suggesting that ectopic PHF7 protein expression in female germ cells potentiates transcription of somatic (gray) genes. On the other hand, the answer may be no—suggesting that one or more of the other 20 SETDB1-controlled genes may be required to silence somatic (i.e. gray) genes. In either case, we do not feel that the answer will substantially add to our main conclusion that SETDB1/H3K9me3 controls female identity via *phf7* transcription.

Figure 4 and legend: The label for Panel B should be *Setdb1*GLKD / WT Ovaries. The term VS. does not give us information about what -5 compared to +5 means. Figure 6 and legend: In the legend for panel C, “regulated” should be “regulate”. Label to Panel D should be *snf148* /WT instead of “VS” , similar as for E. The last sentence is confusing.

Fixed. Thank you for pointing this out. The last sentence now reads: *“Genes that are not normally expressed in ovaries but are ectopically expressed in setdb1 GLKD are labeled in red or in blue. Blue indicates genes which are normally expressed in testis.”*

Page 4, lines 24-26: Isn't NOS expressed again in growing oocytes? Is the NOS-Gal 4 also so expressed? If so, then the statement that NOS-Gal 4 drives expression “largely restricted to pre-meiotic germ cells” would not be correct, at least for ovaries.

Thank you for catching this error. We have removed this line.

Throughout: All the results should be stated in the past tense. Some of examples of corrections needed: page 5 line 6; page 5 lines 10 -11; page 5 line 19; page 6 line 8, etc..

Fixed

Page 5, line 21: Fix wording: should read “...the male-specific transcript...”. Clarify with writing and in figure label and legend whether both the male and female transcripts are produced in the

mutant ovaries, or does the male transcript turn on and the female transcript turn off, as implied by the label of Figure 2 panel A.

Fixed (see note above).

Page 5, lines 25 -26: I thought ovarioles in newly eclosed (unfed) females had egg chambers up to stage 7 (onset of yolk deposition) – thus they contain many cell types not found in germ line tumors. If so, the statement here seems misleading.

Sorry for the confusion. We have removed this line.

Page 6, line 9: The testis specific *phf7* transcript is only slightly apparent in the RNA Seq reads from *hp1a* KD ovaries shown in Figure 2B, so this statement should be modified.

Although the testis-specific *phf7* transcript is only slightly apparent in *hp1a* GLKD ovaries in Fig. 2B, the RT-qPCR data clearly shows that its presence is significantly higher in mutant ovaries than in wild-type ovaries. Furthermore, we now show in Fig. 2e, that PHF7 protein is detectable in *hp1a* GLKD ovaries. We therefore stand by our statement that “*testis-specific phf7 transcript, phf7-RC, is ectopically expressed in setdb1, wde, and hp1a GLKD mutant ovaries.*”

Page 6 bottom and figure 3D: Are the “testis genes” shown in red in panel 3D expressed in testis and elsewhere or are they expressed much higher in testis than in other tissues? Actually, the genes are in all cells, so the wording should be changed here to refer to the transcripts or to expression.

We apologize for not providing enough information regarding definitions in our initial submission. To clarify our meaning, we have revised the RNA-seq analysis section (lines 124-137), beginning with: “*While we found that there was a significant overlap between the genes ectopically expressed in all three mutant backgrounds (Fig. 3c), we did not find that they were enriched for specific gene ontology terms. However, the pivotal role of *phf7* in controlling germ cell sex identity suggested to us that many of the ectopically expressed genes might be normally expressed in testis. To test this idea, we compared our data with published RNA-seq analysis of wild-type testis and *bgn* mutant testis (Shan et al., 2017). In spermatogenesis, *bgn* is required for the undifferentiated spermatogonia to stop mitosis and transition into the spermatocyte stage. In *bgn* mutants this transition is blocked and the testis are enriched for dividing spermatogonial cells. The comparison of the wild-type and mutant expression profiles can therefore be used to identify genes preferentially expressed in early-stage spermatogonia (>2-fold increase in *bgn* compared to wild-type, in blue) and in late-stage spermatocytes (>2-fold decrease in *bgn* compared to wild type, in green). We also identified genes that are normally expressed in testis but are not differentially expressed (**FPKM > 1 in both samples, in red**) and genes that are not detectable in either sample (FPKM <1, in gray).*”

Only after the CHIP analysis, which focused our attention on the core 21 genes, do we differentiate between those genes that are normally *only* expressed in testis and those genes which are normally expressed in testis as well as in other adult tissues (see Table 1).

Page 7 top and figure 3: A good 3rd to 40% of the genes ectopically up regulated in ovaries when *Setdb1* or its partners have been knocked down in germ cells are not normally detected as expressed in testis, indicating that *Setdb1* and H3K9me3 regulate much more in ovaries than the male germ cell specific program.

We elaborate on this excellent point in a new paragraph (lines 141-148), quoted above.

This begs the question of how much of the phenotype observed in ovaries is due to the mis-expression of male germ cell specific programs. What is the effect if the male specific transcript of Phf7 is knocked down in or deleted at the same time as knock down of Setdb1 or its partners?

This is an excellent suggestion. We generated the *phf7^{null}*; *setdb1* GLKD double mutants and report our analysis in Supplementary Fig. 4. Our discussion of the data (lines 175-182) reads: “Previous studies have shown that ectopic PHF7 protein expression is sufficient to disrupt female fate and give rise to a germ cell tumor (Shapiro-Kulnane et al., 2015). We therefore asked if ectopic PHF7 contributes to the *setdb1* GLKD mutant phenotype by generating double mutant females. We found that while loss of *phf7* did not restore oogenesis, there was a shift in the distribution of mutant phenotypes such that the majority of *phf7^{ΔN18}*; *setdb1* GLKD double mutant ovarioles contained no germ cells (Supplementary Fig. 4). While these data indicate that *phf7* is a critical target of SETDB1 silencing, our finding that the phenotype was not fully rescued suggests that ectopic expression of one or more of the other SETDB1 target genes we identified in this study also contribute to the tumor phenotype.”

We note that we are not surprised that the double mutant females were not fertile because SETDB1 clearly does more than regulating *phf7* transcription. But, we did see that the phenotype was nudged towards a less tumorous phenotype indicating that *phf7* plays a role in tumorigenesis.

Page 7, lines 12 -15, and elsewhere: The authors should fill the readers in on whether there are other enzymes in *Drosophila* that can make the H3K9me3 chromatin mark, and what role these may play.

As suggested, we now begin the results section (lines 68-69) with the following statement: “Of the three *Drosophila* enzymes known to methylate H3K9, only SETDB1 is required for germline development (Elgin and Reuter, 2013).”

Page 7, line 17 - 22: The enrichment by ChIP for of H3K9me3 over the *phf7* – RC promoter region shown in figure 4B is not lost – but reduced. The authors should change the wording here accordingly.

Fixed

Figure 6: It would be useful to show additional panels with black and white images of the HA – SetDB1 signal alone in figure 6B as well, so that the readers can more easily see the cytoplasmic distributions without interference from the Vasa signal.

Done. In addition, a Western blot is now included in Fig. 6e and described (lines 214-215): “Western blot analysis of ovarian extracts showed that *snf¹⁴⁸* tumors and wild-type ovaries have a similar level of SETDB1 protein (Fig. 6e, Supplementary Fig. 6).”

Reviewer #4 (Remarks to the Author):

Preserving sexual identity is required for gametogenesis. To do execute this during *Drosophila* oogenesis select testis-enriched genes are repressed. Previously, the authors have show that ectopic expression of a male specific isoform of *phf7* (*phf7*-RC) in the female germ line leads to tumors of undifferentiated cells. In this manuscript the authors propose that SETDB, and pathway members, silence testis genes (such as *phf7*-RC) via H3K9me3 deposition. The authors should address the following questions/suggestions prior to publication.

o In Figure 1A-1C Smolko et al. are trying to show that germ line loss of *setdb1* and *wde* leads to ablation of H3K9me3 mark. But, as the channels are merged it makes it challenging to see this reduction. To make this clearer, please quantitate the levels of H3K9me3 and separate the channels.

Done. The quantitation is provided in Supplementary Fig. 2e.

o In Figure 2 the authors quantitate levels of *phf7*-RC in WT testis, WT ovaries, *setdb1*, *wde*, and *hp1a* GLKD, showing an increase in levels of *phf7*-RC upon loss of SETDB1, WDE, and HP1a. This finding would be more meaningful if PHF7 protein levels were also increased in *setdb1*, *wde*, and *hp1a* GLKD compared to undifferentiated cells in WT ovaries.

This is an excellent suggestion. We show that PHF7 protein is ectopically expressed upon loss of SETDB1, WDE and HP1a in Fig. 2b-e. The discussion in the text (lines 105-110) reads: *“Next, we asked whether ectopic phf7-RC expression correlates with ectopic PHF7 protein expression. Previous work using an HA-tagged phf7 locus in the context of an 20 kb BAC rescue construct showed that PHF7 protein is normally restricted to testis (Shapiro-Kulnane et al., 2015; Yang et al., 2012). We found that in contrast to wild-type ovaries, HA-PHF7 protein is detectable in the cytoplasm and in the nucleus of setdb1, wde, and hp1a mutant ovaries (Fig. 2b-e).”*

o In Figure 3C the authors show that there are several overlapping targets of *setdb1*, *wde*, and *hp1a* GLKD. Do these genes have an enriched-GO term associated with them?

Unfortunately, no. We incorporate this information in the revised document (lines 124-129): *“While we found that there was a significant overlap between the genes ectopically expressed in all three mutant backgrounds (Fig. 3c), we did not find that they were enriched for specific gene ontology terms. However, the pivotal role of phf7 in controlling germ cell sex identity suggested to us that many of the ectopically expressed genes might be normally expressed in testis. To test this idea, we compared our data with published RNA-seq analysis of wild-type testis and bgcn mutant testis (Shan et al., 2017).”*

o In Figure 6A-B Smolko et al. show that SXL also regulates expression of testis genes during oogenesis. To better show that H3K4me3 levels are reduced and SETDB1 levels are unaffected, please quantitate histone mark and HA levels.

Done. Please see Supplementary Fig. 2e and new panels in Fig. 6 (a”, b”, c” and d”) and the Western blot in Fig. 6e and Supplementary Fig. 6.

Citations:

Chau, J., Kulnane, L. S. and Salz, H. K. (2009). Sex-lethal facilitates the transition from germline stem cell to committed daughter cell in the *Drosophila* ovary. *Genetics* 182, 121–132.

Chau, J., Kulnane, L. S. and Salz, H. K. (2012). Sex-lethal enables germline stem cell differentiation by down-regulating Nanos protein levels during *Drosophila* oogenesis. *Proc. Natl. Acad. Sci. U.S.A.* 109, 9465–9470.

Clough, E., Moon, W., Wang, S., Smith, K. and Hazelrigg, T. (2007). Histone methylation is required for oogenesis in *Drosophila*. *Development* 134, 157–165.

- Clough, E., Tedeschi, T. and Hazelrigg, T. (2014). Epigenetic regulation of oogenesis and germ stem cell maintenance by the *Drosophila* histone methyltransferase Eggless/dSetDB1. *Dev Biol* 388, 181–191.
- Elgin, S. C. R. and Reuter, G. (2013). Position-effect variegation, heterochromatin formation, and gene silencing in *Drosophila*. *Cold Spring Harbor Perspectives in Biology* 5, a017780–a017780.
- Klattenhoff, C., Xi, H., Li, C., Lee, S., Xu, J., Khurana, J. S., Zhang, F., Schultz, N., Koppetsch, B. S., Nowosielska, A., et al. (2009). The *Drosophila* HP1 homolog Rhino is required for transposon silencing and piRNA production by dual-strand clusters. *Cell* 138, 1137–1149.
- Leader, D. P., Krause, S. A., Pandit, A., Davies, S. A. and Dow, J. A. T. (2018). FlyAtlas 2: a new version of the *Drosophila melanogaster* expression atlas with RNA-Seq, miRNA-Seq and sex-specific data. *Nucleic Acids Res* 46, D809–D815.
- Mohn, F., Sienski, G., Handler, D. and Brennecke, J. (2014). The rhino-deadlock-cutoff complex licenses noncanonical transcription of dual-strand piRNA clusters in *Drosophila*. *Cell* 157, 1364–1379.
- Nagengast, A. A., Stitzinger, S. M., Tseng, C.-H., Mount, S. M. and Salz, H. K. (2003). Sex-lethal splicing autoregulation in vivo: interactions between SEX-LETHAL, the U1 snRNP and U2AF underlie male exon skipping. *Development* 130, 463–471.
- Rangan, P., Malone, C. D., Navarro, C., Newbold, S. P., Hayes, P. S., Sachidanandam, R., Hannon, G. J. and Lehmann, R. (2011). piRNA production requires heterochromatin formation in *Drosophila*. *Curr Biol* 21, 1373–1379.
- Shan, L., Wu, C., Chen, D., Hou, L., Li, X., Wang, L., Chu, X., Hou, Y. and Wang, Z. (2017). Regulators of alternative polyadenylation operate at the transition from mitosis to meiosis. *J Genet Genomics* 44, 95–106.
- Shapiro-Kulnane, L., Smolko, A. E. and Salz, H. K. (2015). Maintenance of *Drosophila* germline stem cell sexual identity in oogenesis and tumorigenesis. *Development* 142, 1073–1082.
- Sienski, G., Batki, J., Senti, K.-A., Dönertas, D., Tirian, L., Meixner, K. and Brennecke, J. (2015). Silencio/CG9754 connects the Piwi-piRNA complex to the cellular heterochromatin machinery. *Genes Dev* 29, 2258–2271.
- Volpe, A. M., Horowitz, H., Grafer, C. M., Jackson, S. M. and Berg, C. A. (2001). *Drosophila* rhino encodes a female-specific chromo-domain protein that affects chromosome structure and egg polarity. *Genetics* 159, 1117–1134.
- Wang, X., Pan, L., Wang, S., Zhou, J., McDowell, W., Park, J., Haug, J., Staehling, K., Tang, H. and Xie, T. (2011). Histone H3K9 trimethylase Eggless controls germline stem cell maintenance and differentiation. *PLoS Genet* 7, e1002426–e1002426.
- Yan, D., Neumüller, R. A., Buckner, M., Ayers, K., Li, H., Hu, Y., Yang-Zhou, D., Pan, L., Wang, X., Kelley, C., et al. (2014). A regulatory network of *Drosophila* germline stem cell self-renewal. *Dev Cell* 28, 459–473.

- Yang, S. Y., Baxter, E. M. and van Doren, M. (2012). Phf7 controls male sex determination in the *Drosophila* germline. *Dev Cell* 22, 1041–1051.
- Yu, Y., Gu, J., Jin, Y., Luo, Y., Preall, J. B., Ma, J., Czech, B. and Hannon, G. J. (2015). Panoramix enforces piRNA-dependent cotranscriptional silencing. *Science* 350, 339–342.
- Zhang, Z., Wang, J., Schultz, N., Zhang, F., Parhad, S. S., Tu, S., Vreven, T., Zamore, P. D., Weng, Z. and Theurkauf, W. E. (2014). The HP1 homolog rhino anchors a nuclear complex that suppresses piRNA precursor splicing. *Cell* 157, 1353–1363.

Reviewer #1 (Remarks to the Author):

I have read the revised manuscript by Smolko et al.

The authors have adequately addressed my concerns from the previous review.

I think that this is an exciting study that reports a novel epigenetic mechanism involved in preservation of germ cell sexual identity.

Reviewer #2 (Remarks to the Author):

On the whole it has been improved and the authors have addressed most of the reviewers concerns. There are no further technical concerns about the content and presentation of the manuscript in its current form. However, what is really new in the manuscript is identification of target gene candidates for SETDB1 in the ovary. Three out of four referees asked the authors to address the question of how *phf7* promotes the male program in the background of a female differentiation plan. The results shown in Supplementary Figure 4 indicate that “while these data indicate that *phf7* is a critical target of SETDB1 silencing, our finding that the phenotype was not fully rescued suggests that ectopic expression of one or more of the other SETDB1 target genes we identified in this study also contribute to the tumor phenotype.” This forces the authors to tone down their previous claim that “The H3K9 methyltransferase SETDB1 maintains female identity in *Drosophila* germ cells by repressing expression of key spermatogenesis genes.” Indeed, the authors have changed the title of the paper to read “The H3K9 methyltransferase SETDB1 maintains female identity in *Drosophila* germ cells.” Thus, the impact of the current paper in the field has been weakened.

Reviewer #3 (Remarks to the Author):

The authors have addressed my concerns in the revised manuscript.

This is an interesting and important story.

Reviewer #4 (Remarks to the Author):

The authors have addressed my concerns. The paper is now suitable for publication.

Re: Submission of final version of NCOMMS-18-14164A

We thank the four reviewers for investing the time and effort in reviewing our manuscript. Our responses to the reviewer's comments, in red, are given in a point-by-point manner below.

Reviewer #1 (Remarks to the Author): I have read the revised manuscript by Smolko et al. The authors have adequately addressed my concerns from the previous review. I think that this is an exciting study that reports a novel epigenetic mechanism involved in preservation of germ cell sexual identity.

Thank you.

Reviewer #2 (Remarks to the Author): On the whole it has been improved and the authors have addressed most of the reviewer's concerns. There are no further technical concerns about the content and presentation of the manuscript in its current form. However, what is really new in the manuscript is identification of target gene candidates for SETDB1 in the ovary. Three out of four referees asked the authors to address the question of how *phf7* promotes the male program in the background of a female differentiation plan. The results shown in Supplementary Figure 4 indicate that "while these data indicate that *phf7* is a critical target of SETDB1 silencing, our finding that the phenotype was not fully rescued suggests that ectopic expression of one or more of the other SETDB1 target genes we identified in this study also contribute to the tumor phenotype." This forces the authors to tone down their previous claim that "The H3K9 methyltransferase SETDB1 maintains female identity in *Drosophila* germ cells by repressing expression of key spermatogenesis genes." Indeed, the authors have changed the title of the paper to read "The H3K9 methyltransferase SETDB1 maintains female identity in *Drosophila* germ cells." Thus, the impact of the current paper in the field has been weakened.

Thank you for your comments. However, we shortened the title of the paper to meet the editorial guidelines of *Nature Communications*. While we too were disappointed that *phf7^{null}; setdb1 GLKD* double mutant females were not fertile, we were not surprised by this finding because SETDB1 clearly does more than regulating *phf7* transcription. *phf7*, however, remains a "key spermatogenesis gene" because ectopic PHF7 protein expression in female germ cells is sufficient to disrupt female fate. Thus, we stand by our original title "The H3K9 methyltransferase SETDB1 maintains female identity in *Drosophila* germ cells by repressing expression of key spermatogenesis genes.", and if permitted by the editors would be happy to use it.

Reviewer #3 (Remarks to the Author): The authors have addressed my concerns in the revised manuscript. This is an interesting and important story.

Thank you.

Reviewer #4 (Remarks to the Author): The authors have addressed my concerns. The paper is now suitable for publication.

Thank you.